# REFERRING LAYER DECOMPOSITION

**Fangyi Chen**[*], **Yaojie Shen**[*], **Lu Xu, Ye Yuan**[†]**, Shu Zhang, Yulei Niu, Longyin Wen**
Intelligent Editing Team, Intelligent Creation, ByteDance
{fangyi.cfy, shenyaojie, lu.xu1, ye.yuan.1}@bytedance.com
{shu.zhang1, yulei.niu, longyin.wen}@bytedance.com

## ABSTRACT

Precise, object-aware control over visual content is essential for advanced image editing and compositional generation. Yet, most existing approaches operate on entire images holistically, limiting the ability to isolate and manipulate individual scene elements. In contrast, layered representations, where scenes are explicitly separated into objects, environmental context, and visual effects, provide a more intuitive and structured framework for interpreting and editing visual content. To bridge this gap and enable both compositional understanding and controllable editing, we introduce the Referring Layer Decomposition (RLD) task, which predicts complete RGBA layers from a single RGB image, conditioned on flexible user prompts, such as spatial inputs (*e.g.*, points, boxes, masks), natural language descriptions, or combinations thereof. At the core is the RefLade, a large-scale dataset comprising 1.11M image–layer–prompt triplets produced by our scalable data engine, along with 100K manually curated, high-fidelity layers. Coupled with a perceptually grounded, human-preference-aligned automatic evaluation protocol, RefLade establishes RLD as a well-defined and benchmarkable research task. Building on this foundation, we present RefLayer, a simple baseline designed for prompt-conditioned layer decomposition, achieving high visual fidelity and semantic alignment. Extensive experiments show our approach enables effective training, reliable evaluation, and high-quality image decomposition, while exhibiting strong zero-shot generalization capabilities. The project will be released at https://yaojie-shen.github.io/project/RLD/

## 1 INTRODUCTION

While modern generative models (Sheynin et al., 2023; Huang et al., 2024b; Chang et al., 2023; Mu et al., 2025; Zhou et al., 2025; Wei et al., 2024; Rombach et al., 2022; Podell et al., 2023; Brooks et al., 2023; Qin et al., 2023) excel at synthesizing realistic images, they typically operate on the image as a whole—without explicit representations of objects, structure, or scene components. This makes it difficult to selectively manipulate individual elements, enforce consistency across edits, or maintain semantic alignment with user intent. To compensate, region-based editing techniques (Zhao et al., 2024; Zhang et al., 2023b) are often used, where a localized mask, box, or prompt guides modifications. However, these approaches are inherently limited: they only affect visible pixels, lack awareness of occlusion and object-level semantics. These limitations highlight the need to move beyond flat pixel arrays toward structured, object-centric scene representations where elements can be individually understood, edited, and composed.

Rather than treating an image as a monolithic canvas, *image layer*, a transparent visual unit (typically encoded in RGBA format) that encapsulates an entire object or scene element, is inclusive of both visible (unoccluded) and hidden (occluded) regions. This abstraction parallels layer-based workflows in tools like Photoshop (Adobe Inc., 2023), where users manipulate visual elements as discrete, stackable units. Layered image representation provides a foundation for fine-grained editing, intuitive composition, modular reuse, and enables deeper semantic understanding of complex scenes.

---

[*]Equal contribution.
[†]Corresponding author: ye.yuan.1@bytedance.com

  This work is for academic research purposes only.

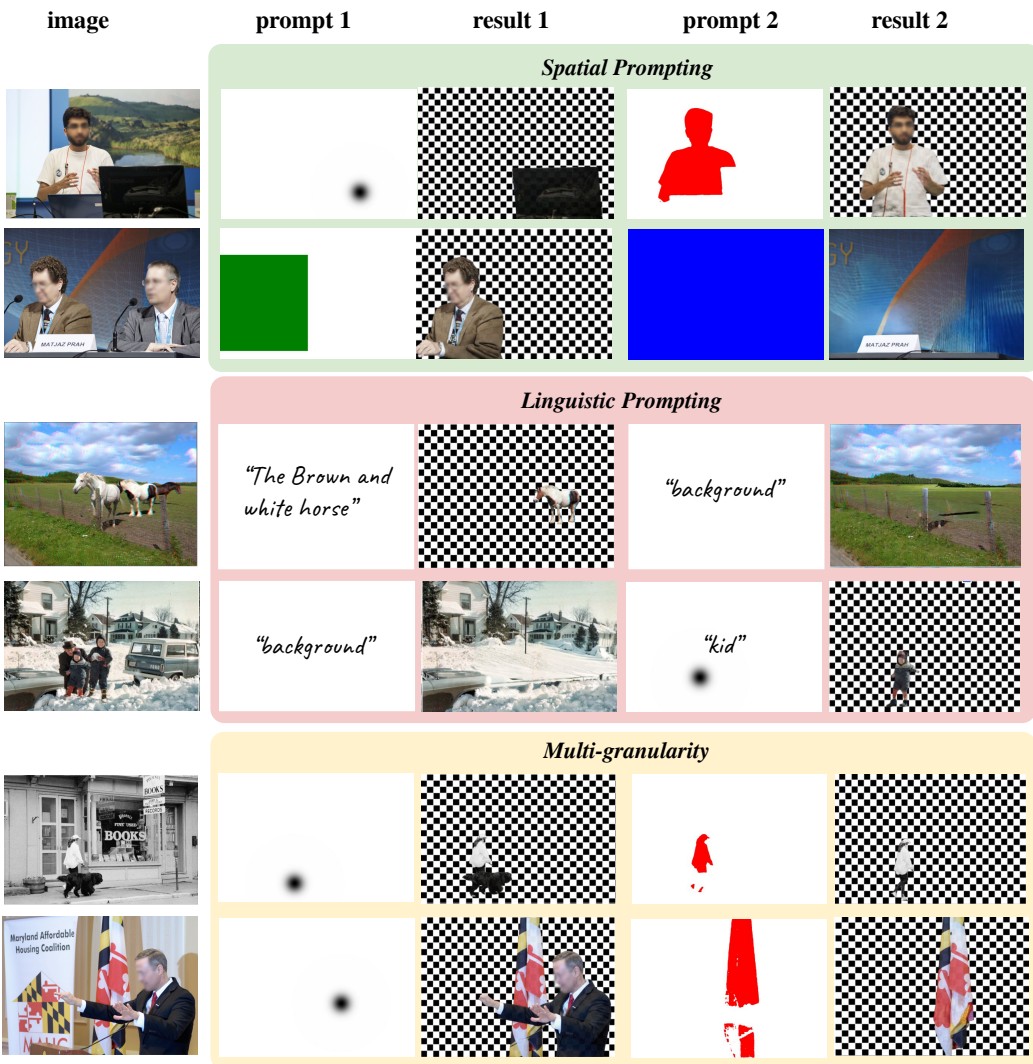

Figure 1: **The RefLayer model trained on RefLade demonstrating the RLD task.** Each row presents two different prompts and their corresponding layer outputs for the same input image. Given an image and diverse user prompts, RLD requires the model to generate targeted and *complete* RGBA layers. The figure showcases examples across prompting modes: Spatial Prompting (e.g., points, boxes, masks) and Linguistic Prompting (e.g., text descriptions like "the brown and white horse" or "background"). Coarse prompts such as a single point may lead to coarse-grained outputs (e.g., a combination of a walker and a dog), while more precise prompts yield accurate, object-specific layers, highlighting multi-granularity capabilities of RefLayer and its strong controllability and generalization.

Recent efforts have begun exploring decomposing images into layers, such as MuLAn (Tudosiu et al., 2024), which decomposes images into layered composites of objects and backgrounds, Text2Layer (Zhang et al., 2023c), which separates an image into two layers (foreground and background), and LayerDecomp (Yang et al., 2024a), which generates RGBA layers from object masks. However, these approaches are constrained by limited data scale and diversity, reliance on synthetic supervision, or coarse compositional definitions. In contrast, real-world applications often require user-controllable access to specific elements, calling for object-centric, prompt-driven layer prediction. We address these needs through scalable data construction, automatic evaluation protocols, and the development of a purpose-built baseline.

We introduce **Referring Layer Decomposition (RLD)**, a novel task that extracts a targeted RGBA layer based on a user-provided prompt (Figure. 1). Prompts can take various forms, including spatial inputs (e.g., points, boxes, masks), linguistic descriptions, or a combination of both. By leveraging this flexibility, RLD allows users to directly specify their object of interest, making it well-suited for applications such as targeted editing, interactive compositing, and object-centric understanding. A central challenge in this task is the lack of large-scale, high-quality training data. To address this, we present **RefLade**, a dataset of 1.11M image-layer-prompt triplets, including 1M auto-generated training examples, 100K manually cleaned layers, and a 10K curated test set—forming the first benchmark for prompt-driven layer decomposition. RefLade is constructed using a scalable data engine that integrates prompt interpretation, RGBA synthesis, and automated filtering, which ensures both quality and extensibility for future data expansion. Building on RefLade, we define an evaluation protocol along three key axes: preservation, completion, and faithfulness, and show that it correlates strongly with human judgments. Together, the RefLade dataset, data engine, and automatic evaluation protocol establish RLD as a trainable, benchmarkable, and researchable task for image decomposition.

Moreover, we propose **RefLayer** to serve as a baseline for the new task. RefLayer is a simple yet effective diffusion-based model that performs prompt-conditioned layer decomposition. It encodes spatial prompts via color-coded maps fused into the latent space, and employs a parallel alpha decoder to predict complete object RGBA layers. We present extensive experiments and ablations on RefLade to validate the evaluation protocol, benchmark the dataset, and assess the baseline model's performance, offering valuable insights for future research.

Our key contributions are as follows:

- We formalize Referring Layer Decomposition (RLD), the first task to explore layer decomposition guided by multi-modal referring inputs.
- We develop a scalable data engine and use it to establish RefLade, a dataset of 1.11M image-layer-prompt triplets with human-curated splits for quality tuning and testing. Along with our designed evaluation protocol, RefLade paves the way for future RLD studies.
- We conduct extensive experiments based on RefLade and benchmark it with a customized baseline model named RefLayer. Experimental results quantitatively and qualitatively validate the effectiveness of the dataset, the evaluation protocol, and the trained model.

## 2 RELATED WORK

**Image Understanding and Editing.** Novel tasks and benchmarks have consistently driven progress in computer vision by enabling breakthroughs and providing standardized ways to measure advancement. The proposed Referring Layer Decomposition intersects with a wide range of tasks, including detection (Ren et al., 2015; Carion et al., 2020; Minderer et al., 2023), segmentation (He et al., 2017; Cheng et al., 2021; Kirillov et al., 2018; Ozguroglu et al., 2024), image generation (Zhang et al., 2023b; Rombach et al., 2022; Ramesh et al., 2022), image editing (Brooks et al., 2023; Kawar et al., 2023; Huang et al., 2024b; Zhang et al., 2024), inpainting (Yu et al., 2018; Lugmayr et al., 2022; Li et al., 2023), and alpha matting (Xu et al., 2017; Park et al., 2022; Yao et al., 2024b). Notably, amodal completion (Xu et al., 2024; Zhan et al., 2020) seeks to infer the appearance of occluded object regions in an image, typically by using amodal segmentation followed by inpainting. While conceptually related, it cannot decompose and output layers. Referring expression segmentation (Hu et al., 2016) segments things based on language descriptions, and promptable segmentation (Kirillov et al., 2023; Ravi et al., 2024) extends it to support a wider variety of prompts to guide multi-granularity segmentation. These tasks are limited to producing masks and do not reconstruct occluded content or support RGBA outputs. In contrast, the proposed RLD is a novel task that unifies and extends these paradigms to generate complete RGBA layers from a single RGB image, conditioned on flexible referring prompts.

**Compositional Image Representations.** The emerging demand in fine-grained image editing and generation has spurred increasing attention towards object-centric and composable image representations. (Schouten et al., 2025; Winter et al., 2024; Mu et al., 2024; Canberk et al., 2024; Wang et al., 2024; Pan et al., 2024) provide object-centric image datasets that focus on editing tasks such as object removal, insertion, repositioning, and resizing. However, these datasets predominantly focus on salient objects and specific editing operations, which offers limited coverage over objects and

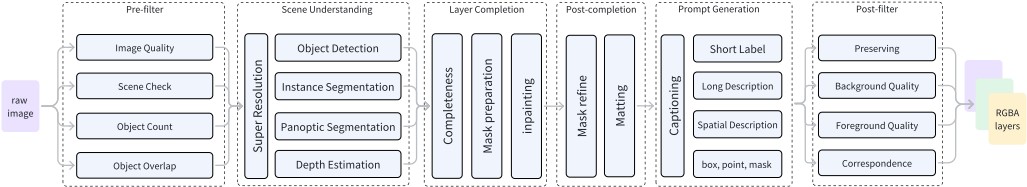

Figure 2: **Overview of the data engine.** The pipeline decomposes a natural image into multiple prompt-aligned RGBA layers through six automatic stages: pre-filtering, scene understanding, layer completion, post-completion, prompt generation, and post-filtering.

spatial relations in real-world images. Recently, several research works have emerged for acquiring RGBA layers in particular. (Tudosiu et al., 2024; Yang et al., 2024a) adopt top-down approaches that decompose RGBA layers from images captured in the wild. Despite high fidelity, these approaches heavily rely on human guidance for both data collection and evaluation, which impairs their scalability. (Zhang et al., 2023c; Zhang & Agrawala, 2024; Sarukkai et al., 2023; Huang et al., 2024a; 2025) attempt bottom-up approaches to synthesize RGBA layers using generative models. However, these methods typically require high-quality RGBA training data, resulting in a circular dependency between model training and data availability. These limitations highlight the urgent need for three unaddressed key building blocks for compositional RGBA layer modeling: a task formulation that flexibly captures granularity in layer semantics, a robust and automated evaluation protocol that eliminates human-in-the-loop reliance, and a data engine that scales up RGBA layer data curation.

## 3 THE REFLADE

This section presents the RefLade dataset (Sec. 3.2), the data engine used to construct it (Sec. 3.1), and the automatic evaluation protocol for benchmarking (Sec. 3.3). Together, these components establish RLD as a trainable and benchmarkable task for layer decomposition research.

### 3.1 DATA ENGINE

Acquiring high-quality RGBA layers with prompt supervision is inherently challenging due to the scarcity of annotated data and the complexity of capturing occluded object appearances. We establish a scalable, modular, and automated data engine designed to get diverse, realistic, and high-fidelity RGBA layers from natural images.

The engine consists of six sequential stages that transform raw natural images into prompt-aligned RGBA layers with complete visual content and semantic grounding, illustrated in Fig. 2: (1) Pre-filter: Screens raw images to ensure they are suitable for decomposition based on quality, content, and object-level structure; (2) Scene Understanding: Detects, segments, and contextualizes salient or "interesting" visual entities that are likely to be user-referenced; (3) Layer Completion: Reconstructs occluded or incomplete object regions to generate visually complete layers; (4) Post-completion: Refines object masks, and predicts alpha mattes; (5) Prompt Generation: Produces diverse referring expressions including spatial, textual, and multimodal prompts that simulate realistic user interactions; (6) Post-filter: Evaluates each RGBA layer's fidelity, realism, and semantic consistency.

More technical details are provided in the Appendix A.1.

The design of the data engine is partially inspired by MuLAn (Tudosiu et al., 2024), but introduces a series of substantial enhancements to lift its success rate (from reported 36% to 70%) as below:

**Efficient Pre-filtering.** Before initiating the computationally expensive generation process, images that are low-quality, visually cluttered, or likely to introduce downstream errors are excluded based on predefined rules. These rules are validated on a task-specific, human-annotated test set of 1000 images, and achieve a decent precision-recall trade-off ensuring that 86.1% of the retained images are suitable for the downstreaming process.

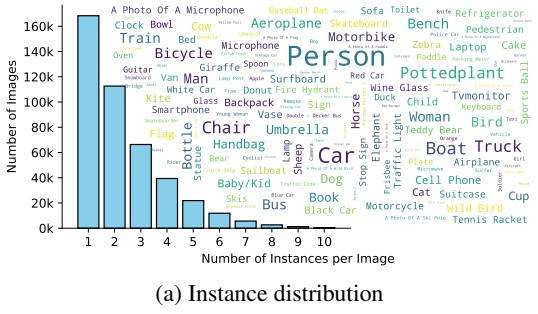
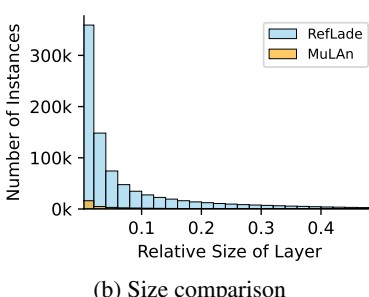

(a) Instance distribution

(b) Size comparison

Figure 3: **Analysis of RefLade dataset.** (a) Instance distribution: Most training images contain 1–3 instances, covering a wide range of object categories. (b) Size comparison: RefLade includes significantly more small instances (by area ratio) than MuLAn.

Table 1: **Comparison of RefLade with related existing datasets.**

| Dataset | Task | # Images | Average Resolutions | # Cls | # Instances | Occlusion Rate | Image Source |
|---|---|---|---|---|---|---|---|
| SAIL-VOS (Hu et al., 2019) | Amodal | 111,654 | 800×1280 | 162 | 1,896,296 | 56.3% | Synthetic |
| OVD (Yan et al., 2019) | Amodal | 34,100 | 500×375 | 196 | - | - | Real |
| WALT (Reddy et al., 2022) | Amodal | 15M | - | 2 | 36M | - | Real |
| AHP (Zhou et al., 2021) | Amodal | 56,599 | - | 1 | 56,599 | - | Real |
| DYCE (Ehsani et al., 2018) | Amodal | 5,500 | 1000×1000 | 79 | 85,975 | 27.7% | Real |
| OMLD (Dhamo et al., 2019) | Amodal | 13,000 | 384×512 | 40 | - | - | Synthetic |
| CSD (Zheng et al., 2021) | Amodal | 11,434 | 512×512 | 40 | 129,336 | 26.3% | Synthetic |
| MuLAn (Tudosiu et al., 2024) | LD | 44,860 | - | 759 | 101,269 | 7.7% | Real |
| RefLade | RLD | 430,488 | 1831×1437 | 12K | 871,829 | 60.8% | Real |

**Enhanced Scene Understanding.** We heavily leverage ensembled strategies in scene understanding. For object detection, the ensemble includes closed-set detection, open-vocabulary detection, and MLLM-based grounding to ensure robustness. For instance segmentation, we ensemble instance segmentation model with panoptic segmentation model to refine instance masks by excluding background masks, which greatly reduces failure in the layer completion stage.

**Prompt Generation.** To enable prompt-driven decomposition and training, we generate a diverse set of referring expressions for each RGBA layer.

**Automated Quality Assurance.** The post-filter enforces quality control by evaluating each RGBA layer along three critical dimensions: (1) Preserving: Assesses alignment between the visible region of the object in the RGBA layer and its counterpart in the original image. (2) Visual Quality: Commercial Vision Language Models is used to evaluate the RGBA layer based on structure, edge quality, and visual realism. (3) Semantic Correspondence: similarity is computed between each RGBA layer and its corresponding caption to verify consistency between textual and visual semantics. Together, these evaluations safeguard the perceptual fidelity and referential accuracy of each output.

**Model Selection.** All models used in our pipeline are chosen based on rigorous internal benchmarking to ensure state-of-the-art performance at the time of submission.

## 3.2 DATASET

To support Referring Layer Decomposition at scale, we construct a large-scale dataset of high-quality RGBA layers paired with diverse prompts, named RefLade. RefLade is generated using the data engine proposed above, and is designed to maximize diversity, realism, and prompt controllability. The statistical comparisons with existing datasets are presented in Tab. 1.

**Dataset Composition.** RefLade consists of 430K images annotated with RGBA layers. On average, each image contributes 2.1 filter-passed foreground instance layers and 0.57 background layers, yielding total of approximately 1.11M RGBA layers. To facilitate prompting, each layer is enriched with bounding boxes and descriptive text in addition to the RGBA channels. To support different

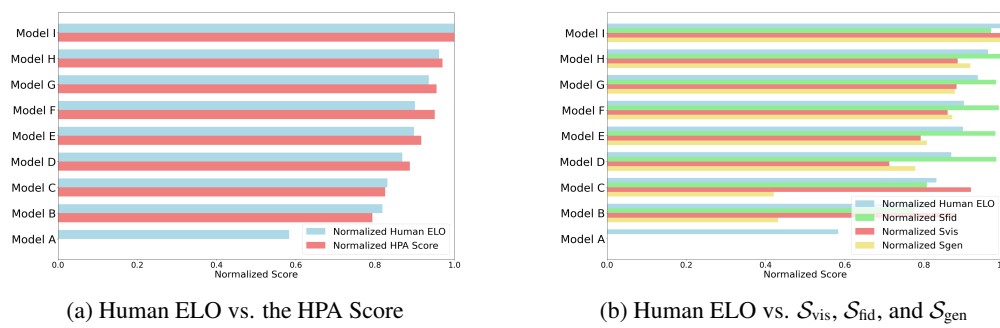

(a) Human ELO vs. the HPA Score

(b) Human ELO vs. $\mathcal{S}_{\text{vis}}$, $\mathcal{S}_{\text{fid}}$, and $\mathcal{S}_{\text{gen}}$

Figure 4: **Comparison of model evaluation metrics.** (a) The HPA score shows strong alignment with Human ELO rankings. (b) In contrast, none of the individual metrics ($S_{\text{vis}}$, $S_{\text{fid}}$, $S_{\text{gen}}$) consistently align with human preferences across models. Model A-I are anonymous for ELO.

stage of model development and evaluation, the dataset is divided into three subsets: a 1M training set, a 100K quality tuning set, and a 10K test set. Notably, both the quality tuning and test sets are manually cleaned to ensure quality.

**Coverage and Diversity.** Images in RefLade are randomly sampled from four public datasets comprising real-world photographs. Fig. 3a and Fig. 3b illustrate the long-tailed distribution of categories, instance counts, and instance sizes per image. RefLade spans a wide spectrum of scenes and objects, including indoor and outdoor environments, vehicles, animals, and diverse human activities. The word cloud in Fig. 3a offers an intuitive overview of the most frequent foreground instance categories, which include "person", "car", "boat", "chair", and "truck" among the top five. Additionally, the dataset is rich in prompt modalities, featuring both sparse spatial cues and detailed textual descriptions.

**Human Annotation.** To complement automated generation, we perform targeted human annotation for quality verification and benchmarking. We randomly sample 92K images, each with associated layers, and recruited 9 professional human annotators to review the samples in terms of background and foreground layer quality. For both background and foreground layers, inpainting completeness, mask accuracy, edge smoothness, and overall realism are evaluated in a "good/neutral/poor" criteria. For foreground instances, 1) a saliency label, defined as whether an object is sufficiently important to warrant its own layer, and 2) an occlusion label, defined as whether an object is occluded by objects in other layers, are annotated in addition. This process took 43 days to complete, yielding a refined subset of 59K high-quality images and a total of 110K validated layers, including 89K foreground and 21K background layers.

**Quality Assessment.** Beyond the curated subset, we conduct an independent quality audit to assess the broader training dataset, where we randomly sample data and instruct annotators to classify each as either "neutral" or "poor". The results show that 74.7% of foreground and 70.2% of background layers meet the quality threshold. These findings validate the effectiveness of our data engine while identifying areas for further improvement.

## 3.3 EVALUATION PROTOCOL

Human judgment of decomposition quality typically considers three aspects: (1) Preservation: how the visible parts of the referring subject is preserved; (2) Completion: how the occluded part of the referring subject is recovered; (3) Faithfulness: how the joint of original visible parts and recovered occlude part is faithful. We aim to establish an evaluation protocol that reflects this multi-faceted human judgment. To this end, we design three assessments that are suitable to reflect each aspect.

**Notation.** The testing dataset $\mathcal{D}$ consists of image-layer-prompt triplets. We denote the (original RGB) image as $i$, the ground-truth RGBA layer as $g$, and a model's prediction towards this testing sample as $p$. A $g$ comprises its RGB channels $g_{\text{rgb}}$, its transparency channel $g_a \in [0, 1]$, and a binary visibility mask $g_v \in \{0, 1\}$ indicating the layer's visible region in the original image $i$. For a background layer, $g_a$ is set to all ones by default, indicating full opacity across the entire region.

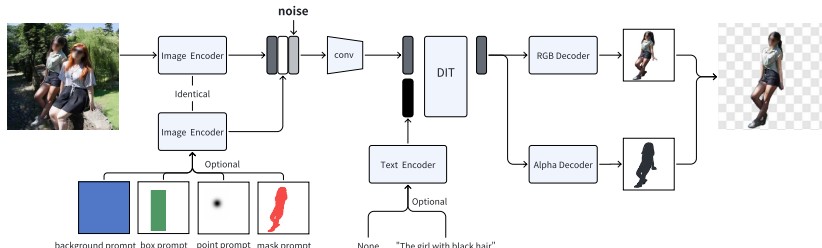

Figure 5: **RefLayer Model architecture.** The model supports prompt-conditioned layer generation using spatial (box, point, mask) and/or textual inputs.

For a foreground layer, we guarantee to provide a high-quality background layer $i_{\text{bkgd}}$ in which the foreground layer $g$ has been cleanly removed.

**Aspect 1: Preservation.** Preserving original visible content is a fundamental requirement in referring layer decomposition (RLD) and is critical for practical use. Before any evaluation, our primary goal is to ensure that the metric focuses on the originally visible regions. Therefore, we first crop $g$ and $p$ using a tight bounding box around the nonzero region of $g_a$, and then mask out regions not visible in the original image based on $g_v$. This preprocessing step ensures that the metric evaluates only the visible content that the model is expected to preserve. We then compute the perceptual similarity using the LPIPS metric (Zhang et al., 2018). Formally,

$$\mathcal{S}_{\text{vis}} = \mathbb{E}_{(p,g)\sim\mathcal{D}}[\text{LPIPS}(g_{\text{rgb}} \odot g_v,\ p_{\text{rgb}} \odot g_v)] \tag{1}$$

where $\odot$ denotes element-wise multiplication. The crop operation is neglected for simplicity.

**Aspect 2: Completion.** Generating reasonable completions is a key feature in RLD. We begin by noting that when a predicted layer successfully completes the occluded regions, the new content should be semantically consistent with that of the ground-truth, even if not pixel-wise identical. To assess this, we introduce *image directional similarity* as a customized metric. Specifically, we extract a CLIP (Radford et al., 2021) feature vector from the visible region of the ground-truth layer $g_{\text{rgb}} \odot g_v$ and compute its directional vector to the (complete) ground-truth layer $g_{\text{rgb}}$. We do the same for the prediction $p_{\text{rgb}}$ in terms of the same origin $g_{\text{rgb}} \odot g_v$. The cosine similarity between directional vectors reflects whether the "action" of completion is similar between ground-truth and prediction. Formally,

$$\mathcal{S}_{\text{gen}} = \mathbb{E}_{(p,g)\sim\mathcal{D}}\left[\cos\left(f(g_{\text{rgb}}) - f(g_{\text{rgb}} \odot g_v),\ f(p_{\text{rgb}}) - f(g_{\text{rgb}} \odot g_v)\right)\right] \tag{2}$$

**Aspect 3: Faithfulness.** We adopt FID to evaluate the distributional similarity between predictions and ground-truth layers. For foreground layers, we first perform alpha blending of the predicted RGBA layer onto the corresponding background image $i_{\text{bkgd}}$, then crop the result using a tight box around the non-zero alpha region to focus evaluation on the relevant content. Formally,

$$\hat{p} = p_{\text{rgb}} \odot p_a + i_{\text{bkgd}} \odot (1 - p_a), \quad \hat{g} = g_{\text{rgb}} \odot g_a + i_{\text{bkgd}} \odot (1 - g_a) \tag{3}$$

$$\mathcal{S}_{\text{fid}} = \text{FID}\left(\{\hat{p} \mid p \in \mathcal{D}\},\ \{\hat{g} \mid g \in \mathcal{D}\}\right) \tag{4}$$

**Aggregating a Unified Score (HPA).** In practice, a unified score combining all is desirable to ease comparison. A challenge is that $\mathcal{S}_{\text{vis}}$, $\mathcal{S}_{\text{gen}}$, and $\mathcal{S}_{\text{fid}}$ differ in both scale and monotonicity. To aggregate meaningfully, we first construct a human-preference ranking using an Elo-based system over 2,000 rounds of pairwise comparisons across 9 models. Guided by the human ranking, we empirically find that applying min-max normalization to each metric and then averaging them yields a score that aligns strongly with human judgments, illustrated in Fig. 4a. We denote the score as the Human Preference Aligned (HPA) score and adopt it as the primary metric for RLD.

## 4 REFLAYER: A BASELINE MODEL

An RLD model is expected to generate an RGBA layer given an image and a variety of prompts. To establish a baseline, in this work, we formulate RLD as a conditional image generation problem. This formulation enables the direct utilization of advanced pretrained models with minimal modifications.

Table 2: **Pearson and Spearman correlations with human ELO across different metrics**

|  | HPA | $S_{vis}$ | $S_{gen}$ | $S_{fid}$ | $S_{vis}+S_{gen}$ | $S_{fid}+S_{gen}$ | $S_{vis}+S_{fid}$ |
|---|---|---|---|---|---|---|---|
| Pearson correlation | 0.96 | 0.90 | 0.96 | 0.94 | 0.95 | 0.95 | 0.94 |
| Spearman correlation | 1 | 0.60 | 0.98 | 0.67 | 0.92 | 0.97 | 1 |

Our model, RefLayer, is illustrated in Fig. 5. Built upon Stable Diffusion 3 (Esser et al., 2024), it uses a VAE encoder (Kingma & Welling, 2013) to encode both the original image and the positional prompt (described below) into latent representations, concatenated channel-wise with a noisy latent vector. A lightweight convolutional layer is applied to compress channels and to align the expected input dimensions of diffusion transformer. The diffusion transformer, conditioned on both the latent and encoded text tokens, conducts the denoising process. After that, two decoders: a standard RGB decoder and a custom alpha decoder are used to reconstruct the RGB content and the alpha transparency mask from the latent.

**Referring Prompts.** RefLayer supports both textual prompts and spatial prompts. While texts are natively supported by most diffusion models, various spatial prompts need a new encoding strategy. We propose unifying all types of positional prompts into colored RGB image format. As seen in Fig. 5, a solid blue canvas for background, a green tightly bounded region for box, a red visible region for mask, and a gaussian heatmap centered at a position for point. This RGB prompt image is encoded into a shared latent space by the same VAE encoder as the original image.

**Alpha Decoder.** The alpha decoder efficiently maps the denoised latent to a transparency mask. Its architecture mirrors that of the VAE decoder, except for the final output channel, whose number is set to one. Functionally, it acts similarly to a matting model but on the latent space, distinguishing it from previous designs (Zhang & Agrawala, 2024; Fontanella et al., 2024; Yang et al., 2024a). This design also isolates the training of the alpha decoder from the overall pipeline, reduces the difficulties of optimization, and keeps the VAE free of change.

**Training.** We freeze the original VAE encoder-decoder, and train the diffusion transformer and the alpha decoder independently, while making sure they share the same latent space. Specifically, the input of the diffusion model is the original image and prompt, and its learning objective is to denoise a decomposed layer that corresponds to the prompt. The alpha decoder's input is the latent embedding of a blended layer, and its learning objective is to decode a mask. After training, the two modules are seamlessly connected. We train the transformer with the standard denoising diffusion loss (Liu et al., 2022; Ho et al., 2020), and the alpha decoder with L1 loss. Check the Appendix A.5 for more details.

## 5 EXPERIMENTS

### 5.1 EVALUATING THE HUMAN PERFERENCE ALIGNED SCORE (HPA)

Fig. 4a compares the HPA scores against the human ELO scores across 9 models spanning a wide range of performances. We observe a remarkably strong agreement between HPA and human ranking, indicating that the proposed metric captures human preferences faithfully. In contrast, Fig. 4b shows that none of the standalone metrics: LPIPS, FID, or image directional similarity (CLIP) consistently aligns with human judgment, each capturing only a partial aspect of human decision. In Tab. 2, we quantitatively verify it via correlations between HPA scores and ELO scores.

To determine the most effective normalization strategy in HPA, we test min-max norm, sigmoid, and log-based scaling. Among these, min-max norm significantly outperforms the alternatives. Because when metrics have large scale gaps, the min-max norm is straightforward that preserves relative differences between models in a linear manner. Particularly, the minimum or maximum bounds of each metric, if uncertain, can be empirically determined from the set of candidate models and reserved for future use, making normalization simple and data-driven.

Table 3: **Benchmarking RefLade with Different Training Set and Scale**. The results are reported on the RefLade testing set, where prompts are given in a multimodal text+box format. RefLadeQ: high-quality tuning set. RefLade+Q: two-stage training of the pretraining plus quality fine-tuning. DIR is the image directional similarity specified in Sec. 3.3.

| Dataset | #layers | Foreground | | | | Background | | | |
|---|---|---|---|---|---|---|---|---|---|
| | | HPA ↑ | FID ↓ | LPIPS ↓ | DIR ↑ | HPA ↑ | FID ↓ | LPIPS ↓ | DIR ↑ |
| MuLAn | 50K | 0.3852 | 22.68 | 0.1403 | 0.2031 | 0.3459 | 21.84 | 0.1588 | 0.6385 |
| RefLade | 50K | 0.4629 | 10.98 | 0.1411 | 0.2543 | 0.5932 | 16.87 | 0.0520 | 0.7206 |
| RefLade | 100K | 0.4621 | 11.27 | 0.1428 | 0.2589 | 0.5935 | 16.73 | 0.0530 | 0.7213 |
| RefLade | 200K | 0.4631 | 10.99 | 0.1434 | 0.2547 | 0.5461 | 19.84 | 0.0552 | 0.6950 |
| RefLade | 400K | 0.4678 | 10.66 | 0.1404 | 0.2575 | 0.5792 | 18.36 | 0.0493 | 0.7129 |
| RefLade | 1M | 0.4685 | 11.10 | 0.1377 | 0.2561 | 0.5587 | 17.35 | 0.0730 | 0.7190 |
| RefLadeQ | 100K | 0.4698 | 10.60 | 0.1378 | 0.2531 | 0.6657 | 12.99 | 0.0487 | 0.7721 |
| RefLade+Q | 1.1M | **0.4813** | 10.50 | 0.1330 | 0.2652 | **0.6682** | 13.14 | 0.0437 | 0.7673 |

Table 4: **Ablation study on Type of Prompts.**

| Prompt | $HPA_{frgd}$ | $HPA_{occ}$ |
|---|---|---|
| Text | 0.2403 | 0.1704 |
| Point | 0.4394 | 0.3530 |
| Box | 0.4719 | 0.4065 |
| Mask | **0.4842** | 0.4270 |
| Text+Mask | 0.4833 | **0.4403** |

Table 5: **Ablation study on Base Model and Background Canvas Color.**

| Base Model | Canvas | $HPA_{frgd}$ | $HPA_{occ}$ |
|---|---|---|---|
| SD3 | Board | 0.4275 | 0.3692 |
| InstP2P | Board | 0.4534 | 0.4100 |
| UltraEdit | Board | **0.4698** | **0.4187** |
| UltraEdit | Black | 0.4607 | 0.4115 |

## 5.2 BENCHMARKING REFERRING LAYER DECOMPOSITION

As RefLade provides the first benchmark of layer decomposition, we use RefLayer as a baseline to benchmark it under a variety of training settings. Without otherwise specified, prompts are given in a multimodal text+box format, which offers sufficient guidance. Although a mask prompt could lead to more accurate localization, box prompts are more realistic for user-facing applications.

Tab. 3 summarizes the effect of training source and scale. Models trained on RefLade consistently outperform the MuLAn-trained baseline, even when trained on the same number of layers, highlighting the superior data quality of RefLade. Foreground performance steadily improves as data scales up from 50K to 1M, indicating that decomposition quality benefits most from more data. In contrast, background performance does not follow a monotonic trend. This reflects that background decomposition primarily involves removing foreground objects, which can be done with fewer training samples of higher quality. In summary: **Foreground decomposition is data-hungry, relying on large-scale learning. While background decomposition benefits more from data quality, the task is more reliant on accurate copy-paste from context than novel generation.**

The model trained on RefLadeQ, the high-quality tuning set, shows a significant boost in background performance. The best overall results are achieved by RefLade+Q, which combines large-scale pretraining with high-quality fine-tuning. This configuration yields substantial improvements across all metrics, demonstrating the effectiveness of scaling both data quantity and quality.

## 5.3 REFLAYER ABLATION STUDY

In Tab. 4, we compare the foreground performance of RefLayer when using different prompt types. We report results on foreground layer ($HPA_{frgd}$) and foreground with occlusion ($HPA_{occ}$). Spatial prompts (point, box, mask) consistently outperform pure textual prompts, indicating that the current model's weakness on text-based localization. Point prompts yield moderate performance (0.4394 / 0.3530). Box prompts achieve a notable improvement (0.4719 / 0.4065), suggesting that coarse spatial localization significantly helps. Mask prompts lead to the highest performance among single-modality inputs (0.4842 / 0.4270), highlighting the advantage of precise spatial constraints. Interestingly, the text+mask prompt achieves a slightly lower $HPA_{frgd}$ (0.4833) compared to mask alone (0.4842), but a significantly higher $HPA_{occ}$ (0.4403 vs. 0.4270). We verified that incorporating text slightly degrades

the model's ability to preserve content within the mask, but enhances its generative capability in occluded regions by providing additional semantic context. We leave this trade-off to future work.

In Tab. 5, we validate that initializing with a well-trained image editing model improves performance compared to using a vanilla text-to-image generation model. Specifically, models based on UltraEdit (Zhao et al., 2024) significantly outperform those based on SD3 (Esser et al., 2024) and InstP2P (Brooks et al., 2023) across all HPAs. Additionally, we conduct an ablation on the selection of background used for blending RGBA layers during training. We find that using a pure color background (e.g., black) can lead the alpha decoder to overfit to trivial cues, resulting in suboptimal performance. In contrast, using a checkerboard-style background with slight color jittering (denoted as "Board") leads to more robust alpha prediction.

## 5.4 HUMAN EVALUATION

We further evaluate RefLayer with the Passrate@K metric. For each test sample, the model generates $K$ results. Human annotators then assess whether at least one of the $K$ outputs is satisfactory. Using this metric, the best model trained on RefLade+Q achieves a Passrate@K=1,5,10 of 28%, 65%, 74% for background layers and 45%, 74%, 79% for foreground layers, respectively.

## 6 CONCLUSION

In this paper, we introduce the task of Referring Layer Decomposition, challenging the extraction of complete, object-aware RGBA layers based on diverse user prompts. To facilitate research in this area, we build a data engine and present RefLade dataset, alongside an evaluation protocol that aligns with human preference. We also propose RefLayer as a baseline. Our work paves the way for layer decomposition research.

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

## A  ADDITIONAL TECHNICAL DETAILS AND REPRODUCIBILITY

### A.1  DATA ENGINE

**Pre-filter.** The pipeline begins with filtering raw image inputs to remove those unlikely to produce useful decompositions. Low-quality or visually cluttered images can propagate errors downstream, so we apply four heuristics: (1) Image quality filters out blurry or poorly lit images; (2) Scene filtering removes inappropriate or meaningless content; (3) Object count restricts samples to images with a manageable number (1–20) of objects; and (4) Object overlap discards images with excessive object congestion. These filters are implemented using in-house aesthetic and content classifiers, along with object detectors RT-DETR (Lv et al., 2024) (closed-set) and OWL-V2 (Minderer et al., 2023) (open-vocabulary).

**Scene Understanding.** After filtering, the pipeline performs semantic and structural analysis of the scene. Our goal is to extract "interesting" objects that are visually salient and likely to be edited by users. To achieve this, we construct a detection ensemble consisting of RT-DETR (Lv et al., 2024), OWL-V2 (Minderer et al., 2023), and a hybrid tagging-grounding system combining GPT-4o (Hurst et al., 2024) with Grounding-DINO (Liu et al., 2023), and annotate a dedicated test-set to guide the design of "interesting object detection", where annotators are asked to label those objects that could potentially raise their attention. Thresholds for detection confidence and bounding box area are tuned to balance precision and recall. Detected objects are subsequently segmented with SAM-V2 (Ravi et al., 2024), and low-resolution images are enhanced using a super-resolution model (Rombach et al., 2022) to preserve detail. In parallel, we extract panoptic segmentation and monocular depth maps using OpenSeeD (Zhang et al., 2023a) and Depth Anything V2 (Yang et al., 2024b), providing essential cues for inpainting and mask refinement in later stages.

**Layer Completion.** With a structured scene representation established, the next step involves identifying and recovering visual content that may be partially occluded. Determining whether an object is visually incomplete is a challenging task, given the complexity of spatial arrangements and inherent visual ambiguity. To handle this, we employ Gemini-2.0 (Google, 2024) with in-context learning to evaluate object completeness. For each instance, when occlusion is detected, we create an inpainting mask and utilize a state-of-the-art generative model (Bria AI, 2024) to reconstruct the missing regions. Specifically, we utilize depth estimation results to construct inpainting masks: based on the average depth of the current instance, we determine a depth threshold, and regions with higher depth values (i.e., likely occluders) are identified and masked for inpainting. Additionally, we exclude background regions (e.g., road, grassland), as identified by panoptic segmentation, to ensure that only foreground occluders are included in inpainting masks. Finally, the inpainting model is guided using both the generated mask and the object's class label to recover the occluded regions. This process ensures that each RGBA layer represents a complete object, encompassing both visible and hidden parts.

**Post-completion.** After inpainting, we refine the object's segmentation to accurately reflect both the original visible regions and the newly reconstructed content. This refinement is performed using SAM-V2, which ensures that the mask tightly encloses the full extent of the object. Next, a high-resolution matting model (Yao et al., 2024a) is applied to predict the alpha channel, capturing fine details along object boundaries and soft transitions.

**Prompt Generation.** To enable prompt-driven decomposition and training, we generate a diverse set of referring expressions for each RGBA layer. These prompts fall into two categories and can be combined: (1) Spatial prompts: Coordinates, bounding boxes, and segmentation masks derived from detection and segmentation. (2) Semantic prompts: we generate descriptive text using GPT-4o (Hurst et al., 2024), which includes, long-form captions that offer comprehensive descriptions, concise labels suitable for prompt-based referencing, and spatial description that reflect that reflect real-world referring behavior (e.g., "the red car on the left side").

**Post-filter.** The final stage of the pipeline enforces quality control by evaluating each RGBA layer along three critical dimensions: (1) Preserving: Assesses alignment between the visible region of the object in the RGBA layer and its counterpart in the original image. Significant discrepancies indicate a likely failure. (2) Visual Quality: Gemini-2.0 (Google, 2024) is used to evaluate the RGBA layer (which is rendered on a checkerboard background), scoring it from 1 to 5 based on structure, edge quality, and visual realism. Layers with scores below the threshold are discarded.

(3) Semantic Correspondence: CLIP similarity is computed between each RGBA layer and its corresponding caption to verify consistency between textual and visual semantics. Together, these evaluations safeguard the perceptual fidelity and referential accuracy of each output, ensuring that only high-quality, semantically meaningful RGBA layers are preserved for downstream use.

## A.2 SOME EXPERIMENT RESULTS ON DATA ENGINE

**Scene understanding.** We use an ensemble of state-of-the-art models (detailed in the paper) to extract diverse information. The ensembled detector achieves 0.652 precision and 0.659 recall, which is better than only using OWL-V2 (0.236 / 0.621) or RT-DETR (0.646 / 0.631).

**Layer completion.** To avoid over-completion errors, we first use Gemini-2.0 to filter out complete objects, which are excluded from further processing in this stage. Gemini-2.0 achieves high precision ( 0.9) in identifying complete objects, allowing us to eliminate  60% of samples that don't require completion, significantly reducing error rates. The remaining 40% proceed to the inpainting stage, though this does not necessarily result in incorrect completions. For these images, we apply an inpainting model to restore the occluded regions. A key challenge is generating effective inpainting masks for occluded areas, which is related to amodal segmentation. To simplify the process, following MuLAn, we generate inpainting masks with high recall relative to the actual occlusion. Specifically, our method filters out neighboring regions with high depth (likely occluders) and excludes background using refined panoptic segmentation. These components form the final inpainting mask. The inpainting model produces an RGB image—a completed object on a white background—without an explicit object boundary. The next stage generates the alpha channel to produce the final RGBA layer.

## A.3 THE ROLES OF MODERN MULTIMODAL LARGE LANGUAGE MODELS

Modern multimodal large language models play a critical role in our proposed data engine for layer decomposition. They serve as both executors and supervisors at key stages of the pipeline. By leveraging their zero-shot generalization and reasoning abilities, we significantly improve the quality of data generation.

**GPT as a Tagging Model.** In the scene understanding stage, identifying all interesting objects is crucial for both foreground and background decomposition. We use GPT-4o (OpenAI, 2024) as a tagging model to detect notable and meaningful objects in the scene. These textual tags are then grounded into spatial regions using a visual grounding model (Grounding-DINO) to produce bounding boxes.

**GPT for Layer Captioning.** For each RGBA layer, we prompt GPT to generate rich, descriptive captions. These include both visual attributes (e.g., "a red double-decker bus on the road") and positional references (e.g., "the chicken on the left").

**Gemini for Judging Object Completeness.** Determining whether an object is fully visible or occluded is critical for choosing the appropriate extraction strategy. If an object is occluded, we invoke the completion stage; otherwise, we directly extract it using a matting-refined segmentation mask, which avoids potential artifacts from generative completion. However, assessing object completeness from an image is nontrivial. Depth-based methods have limitations (e.g., sensitivity to viewpoint or object class). To address this, we leverage Gemini-2.0's (Google, 2024) in-context visual reasoning abilities. Prompt engineering is illustrated in Fig. 6. Gemini-2.0 achieves superior performance on this task compared to GPT-4o, reaching 90.3% precision and 56.1% recall versus GPT-4o's 66.7% precision and only 0.08% recall at the time of this submission.

**Gemini for Quality Control.** Gemini also plays a key role in dataset quality control. We use targeted prompts to evaluate both foreground and background layers. For foregrounds, we assess structural completeness, edge integrity, and realism. For backgrounds, we check object removal effectiveness, absence of unintended artifacts, and visual plausibility. Through empirical comparison, we find Gemini-2.0 to be more reliable than GPT-4o in tasks requiring fine-grained inspection of visual details.

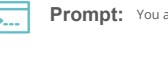 **Prompt:** You are given two image pairs, the first pair is an example, the second pair needs your judge . Each pair has two images placed side by side, where:
1. The left image is the original Image.
2. The right image is the target object segmented from the left image.

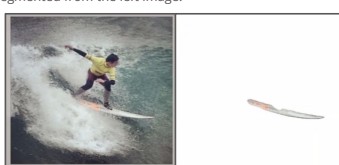

For example, in the example pair, the object in the right image is a surfboard. In the left image, part of the surfboard is occluded by water and also occluded by the man standing on it. Thus, the target object (surfboard) is not complete."

Your task is to judge the (second given) testing pair whether the target object has not been occluded by anything in the original image. During the process, you should
1. Take a close look at the target object in the right image.
2. Refer to the left image, verify if there is anything that is in front of the target object and has occluded it,
3. Also verify whether potential environments (river, glass, snowland, etc) has blocked parts of the object,
4. Return your answer as Yes or No and reason.

**Testing Image:** **w/o zoom in:**                                     **w/ zoom in:**

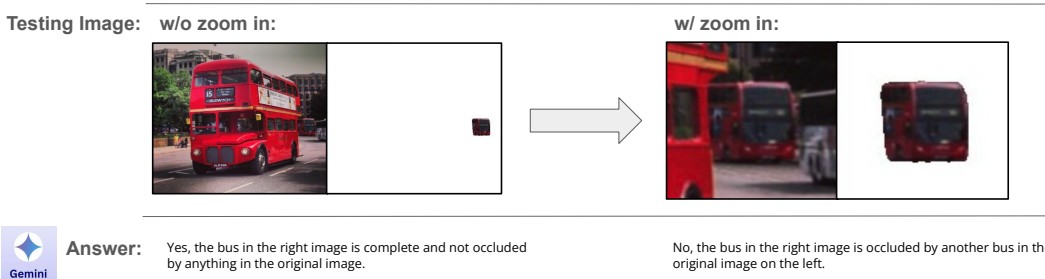

**Answer:** Yes, the bus in the right image is complete and not occluded by anything in the original image.        No, the bus in the right image is occluded by another bus in the original image on the left.

Figure 6: **Prompt engineering with gemini-2.0 for judging completeness.**

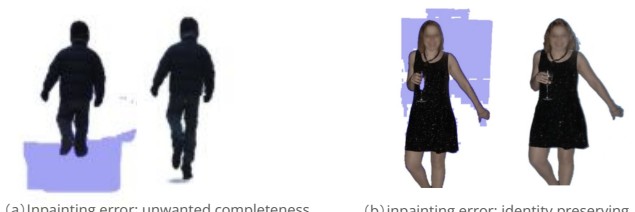

(a) Inpainting error: unwanted completeness      (b) inpainting error: identity preserving

Figure 7: **Data Engine Inpainting Error Example.** (a) The person is completed with extra length of leg. (b) The person's identity is not preserved after processing.

## A.4 ERROR SOURCE ANALYSIS

Given the complexity of the task, our data engine is designed as a six-stage sequential pipeline, where each stage may consist of multiple foundation models running in parallel or in series. Errors introduced in earlier stages may be cascaded downstream, either causing failures or being partially mitigated by later stages. Here, we analyze two primary sources of error observed in the data engine.

**Inpainting Errors (∼65% of Total Errors).** Inpainting in the layer completion stage is critical to the success of the data engine. It relies on a carefully defined inpainting mask, which should cover occluders in front of the target object, typically inferred from depth cues and objects detected. However, in complex scenes, the mask may include regions that should not be modified. This can lead to undesired hallucinations or visual artifacts (Fig. 7 (a)).

Additionally, the inpainting model itself may fail to preserve identity or produce low-quality completions (Fig. 7 (b)). With the rapid progress in generative models, we expect future improvements in inpainting performance to mitigate such issues.

**Segmentation Error (∼20% of Total Errors).** Segmentation model SAM is employed in both the scene understanding and post-completion stages. Although SAM is generally robust, it can still

produce incomplete masks or over-segment regions, occasionally including irrelevant objects or omitting important parts of the target.

**Other Error Source.** Beyond the two major sources above, additional errors may arise from missed detections, inaccurate panoptic segmentation, completeness judgment, or matting artifacts.

### A.5 RefLayer Model Training

The training of Transformer follows the standard latent diffusion training: We add Gaussian noise to the target latent at a random timestep to produce a noisy latent, and train to predict the added noise, so the learning objective is to minimize the mean squared error (MSE) between the predicted and actual noise. The target latent is manufactured from the desired RGBA layer: we first alpha-blend the RGBA onto a standard gray-white checkerboard pattern, then encode it by VAE into the target latent (Sec. 7.3). The training uses AdamW with a constant learning rate 5e-5. On RefLade 1M, the training uses 64xA100 GPUs with batch size 1024 for 7k steps, and on the other RefLade splits, we use 8xA100 GPUs with batch size 256 for 3.5k steps. The alpha decoder is trained to predict an alpha mask directly from the target latent. We optimize by minimizing the L1 distance, with learning rate 1e-4 for 7k steps on RefLade 1M.

## B Limitation

- We ignore the effects of shadows, reflections, rain, dust, and similar factors. While decomposition with these effects could lead to new problems (e.g., a shadow in one scene may not be appropriate when transferred to another with different lighting conditions). Nevertheless, despite its complexity, such side effects should be considered in future work.

- Inevitably, our dataset contains some noise that is difficult to eliminate, arising from the accumulation of errors across different stages of the data engine. Although models are capable of learning effectively even when trained on noisy data, we hope that the data quality could be improved with the development of better foundation models.

- Mutual occlusion could be a problem when composing a scene from multiple decomposed layers; in this case, a visibility mask for each layer should be properly given either by a human or a trained model. Since our proposed task aims to decompose rather than compose, we neglect the problem at this stage.

- Our proposed baseline model has not yet been evaluated using more advanced architectures. We anticipate that exploring stronger models could lead to significant performance gains and leave this as an open direction for future research.

- The current data generation pipeline operates primarily at the instance level. It does not account for finer-level of granularity, such as parts of object. Extending the pipeline to incorporate parts understanding would further enhance it.

## C Discussion

### C.1 Concrete applications in the future

The flexibility enabled by multi-modal referring unlocks a broad spectrum of potential applications in real-world products. Here are several representative use cases:

- Full-Scene Decomposition Agent: With modern MLLMs exhibiting strong scene understanding, it becomes feasible to prompt them to describe an image and generate bounding boxes for each object. RefLayer can then be applied to decompose each identified region into high-quality RGBA layers.

- Scalable RGBA Image Generation: RefLayer offers a reliable and scalable solution for producing RGBA images from diverse prompts, making it a valuable tool for generating training data at scale.

- Seamless Integration with Existing Tasks: RefLayer naturally complements and enhances a variety of existing tasks, such as amodal segmentation, object completion, foreground removal, and layered image editing.

- The RefLade dataset could potentially support finetuning/data augmentation for a variety of generative models on object removal, insertion, repositioning, resizing, and image editing tasks.

### C.2  WITH THE DATA ENGINE, DO WE STILL NEED A UNIFIED MODEL?

The development of such a data engine naturally leads to an important question: if the engine is capable of performing the Referring Layer Decomposition task, does a unified model still need to be trained? The answer remains yes, for two main reasons.

First, the data engine is computationally heavy, time-consuming, and expensive. Due to the reliance on multiple pretrained models and commercial MLLMs, each decomposition takes approximately two minutes, making it impractical for real-time or large-scale deployment.

Second, while the data engine produces noisy outputs, it can generate data at scale with no human supervision. With sufficient training data and robust optimization strategies, a unified model trained on this noisy dataset has the potential to learn generalizable patterns, smooth over noise, and ultimately outperform the data engine that generated its training data.

### C.3  RATIONALE FOR THE MIN-MAX NORMALIZATION.

The purpose of applying normalization is to bring all metrics onto a comparable scale, as they inherently differ in range and magnitude. Specifically, FID ($S_{fid}$) and LPIPS ($S_{vis}$) have a known theoretical lower bound of 0 but no fixed upper limit, while CLIP directional similarity ($S_{gen}$) is bounded above by 1. Given these characteristics, we find min-max normalization to be the most suitable approach compared to other normalization methods for the following reasons:

1. Alignment with Theoretical Bounds: Min-max normalization inherently respects the known theoretical bounds of each metric, offering a stable and interpretable frame of reference for future studies. In contrast, methods like z-normalization or MAD normalization disregard these bounds, instead centering the scores around dataset-dependent statistics. This can introduce bias tied to the distribution of ELO-evaluated models rather than the metrics' theoretical properties.

2. Robust Definition of Lower Bound: Min-max normalization only requires selecting a representative "low-performing" model to define the lower end of the scale. Since our ELO model pool is diverse and spans a broad spectrum of performance levels, this selection avoids the outlier issue and ensures a meaningful range.

3. Interpretability and Practicality: Normalizing within a bounded range assumes a linear relationship between metric scores and human preferences. While this linearity may not be perfect, it is a reasonable and interpretable approximation. More complex nonlinear mappings would require extensive ELO-style human studies, which are impractical at scale.

## D  ADDITIONAL EXPERIMENTS WITH AMODAL SEGMENTATION & COMPLETION

### D.1  RLD VS. AMODAL SEGMENTATION & COMPLETION

Previous works on amodal segmentation has focused on predicting the mask of an object, including its occluded parts. Building on this, amodal completion goes a step further by aiming to infer the complete appearance of the occluded regions. Our proposed RLD task inherently encompasses both amodal segmentation and completion capabilities, while it introduces fundamental differences. We emphasize them as follows.

**Input Modality.**  Amodal segmentation and completion typically require an input image and a binary mask indicating the target object. In contrast, RLD is designed to support a broader range of prompt types, including both spatial prompts and textual prompts, making it a multimodal task that enables greater flexibility.

**Output Representation.**  The outputs of amodal segmentation are binary instance masks that extend beyond the visible regions. Amodal completion focuses on generating de-occluded image content.

Neither task is designed to produce a complete RGBA layer, which is central to RLD. The ability to generate such composable layers is what fundamentally distinguishes RLD from prior work.

**Modeling.** Typical amodal completion approaches, such as Pix2Gestalt (Ozguroglu et al., 2024), adopt a two-step pipeline: first employ a diffusion model for inpainting or conditional generation, followed by a separate matting model to obtain the amodal mask. These steps are often handled by isolated models. In contrast, RefLayer is an end-to-end, unified model, making it simple and fast.

**Task Definition and Benchmarking.** To the best of our knowledge, amodal completion lacks a standardized task formulation, benchmark, and automatic evaluation. In contrast, RLD is defined as a clear, prompt-driven decomposition task. We establish a reliable benchmark for automatic quantitative evaluation, grounded in human preference alignment, which will accelerate the research in the field.

Since RLD and the trained RefLayer model support amodal segmentation and completion, we present additional experiments below to validate their performance in these contexts.

### D.2 AMODAL SEGMENTATION EXPERIMENTS.

We use the COCOA dataset (Zhu et al., 2017), a widely used benchmark for amodal segmentation. We exam our RefLayer with "text+mask" prompt trained on RefLade, and conduct zero-shot evaluation on COCOA. Specifically, given the visible mask and text label of an occluded object as inputs, we generate the complete object representation and extract the alpha channel from the output as the predicted amodal mask. While the RefLayer is a diffusion model and is able to produce diverse outputs, we opt for a single prediction per object to account for the computational cost of diffusion models. Following prior works (Gao et al., 2023; Ozguroglu et al., 2024), we use $mIoU_{full}$ and $mIoU_{occ}$ to evaluate the quality of the amodal masks, where $mIoU_{full}$ measures the mean IoU of full object mask, and $mIoU_{occ}$ measures the mean IoU between occluded regions. As shown in Tab. 6, without bells and whistles, RefLayer achieves state-of-the-art result on $mIoU_{occ}$, outperforming the other methods by a significant margin, proving its ability in occluded completion. Please note that RefLayer is never trained on COCOA, verifying its zero-shot generalization. RefLayer also achieves a competitive result on $mIoU_{full}$.

It is worth noting that even a small improvement in HPA on RefLade can lead to a significant gain in mIoU on downstream task such as COCOA. As shown in Tab. 6, RefLayer-$\gamma$ achieves 0.0059 higher HPA than RefLayer-$\alpha$, with improvements of 1.76 in $mIoU_{full}$ and 6.52 in $mIoU_{occ}$ on the COCOA dataset. This highlights that RLD is a challenging task, and HPA is a strict metric considering three aspects of preservation, completion, and faithfulness.

### D.3 AMODAL COMPLETION EXPERIMENTS.

To the best of our knowledge, there is no other public benchmark for amodal completion. We evaluate amodal completion on the test subset of our RefLade dataset, and select Pix2Gestalt (Ozguroglu et al., 2024), which is a recent work on amodal segmentation and completion. Pix2Gestalt is trained on its own synthetic data. As shown in Tab. 7, RefLayer outperforms Pix2Gestalt in terms of all metrics. This is because of the data quality between the RefLade and Pix2Gestalt's synthesis data, and the model differences between them.

## E QUALITATIVE RESULTS OF THE REFLAYER MODEL

### E.1 TEXT-ONLY PROMPTING IS MORE CHALLENGING THAN SPATIAL PROMPTING

As observed in quantitative evaluation, models conditioned solely on textual prompts generally achieve lower performance metrics compared to those guided by explicit spatial inputs. This disparity arises from the intrinsic challenges text-only prompting introduces to the underlying vision model's localization and reasoning capabilities.

We visualize this challenge in Fig. 8. The text-only prompt inherently requires the model to perform an additional reasoning and localization step to correctly identify the target object. For instance, given

Table 6: **Comparison with amodal segmentation methods on COCOA.** RefLayer-$\alpha$, $\beta$, and $\gamma$ denote our model trained on 100K high-quality set, 1M training set, and both sets, respectively, using text+mask prompts. The * denotes that we reproduce the results based on a single inference attempt using the official checkpoint provided by the authors.

| Methods | HPA$_{frgd}$ ↑ | Zero-Shot | mIoU$_{full}$ | mIoU$_{occ}$ |
|---|---|---|---|---|
| PCNet (Zhan et al., 2020) | - | ✗ | 76.91 | 20.34 |
| VRSP (Xiao et al., 2021) | - | ✗ | 78.98 | 22.92 |
| AISformer (Tran et al., 2022) | - | ✗ | 72.69 | 13.75 |
| C2F-Seg (Gao et al., 2023) | - | ✗ | **80.28** | 27.71 |
| Pix2Gestalt* (Ozguroglu et al., 2024) | 0.3397 | ✓ | 70.02 | 26.79 |
| RefLayer-$\alpha$ | 0.4833 | ✓ | 73.28 | 32.31 |
| RefLayer-$\beta$ | 0.4829 | ✓ | 72.70 | 34.85 |
| RefLayer-$\gamma$ | **0.4892** | ✓ | 75.04 | **38.83** |

Table 7: **Comparison with amodal completion methods on RefLade test subset.** RefLayer-$\alpha$ denotes the same model as in Tab. 6.

| Method | HPA$_{occ}$ ↑ | Foreground | | | |
|---|---|---|---|---|---|
| | | HPA ↑ | FID ↓ | LPIPS ↓ | DIR ↑ |
| Pix2Gestalt (Ozguroglu et al., 2024) | 0.2687 | 0.3397 | 18.23 | 0.1671 | 0.1198 |
| MuLAn (Tudosiu et al., 2024) | 0.3041 | 0.3852 | 22.68 | 0.1403 | 0.2031 |
| RefLayer-$\alpha$ | **0.4403** | **0.4833** | **10.43** | **0.1298** | **0.2586** |

the prompt "two red barns" (top row), the model incorrectly extracts a large portion of the foreground ground plane and horses, demonstrating a failure to isolate the target based on the language alone. In contrast, a simple Spatial Prompt (e.g., a green box) provides an explicit localization cue, guiding the model to accurately decompose the target object(s). The prompt "person standing in the middle" (bottom row) similarly yields a poor result without spatial guidance, but a generic box yields a high-quality layer extraction.

A promising approach to bridge this gap could involve applying a grounding model to provide additional spatial prompting cues when running RefLayer in text-only mode. Alternatively, a unified and more powerful initialization, possessing strong localization and reasoning ability, could also significantly improve performance.

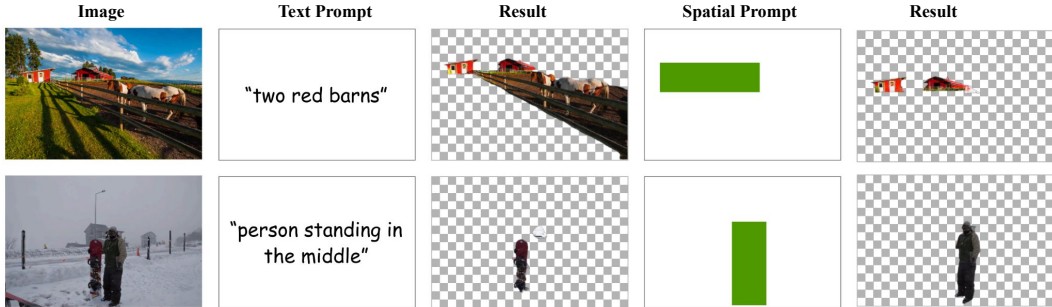

Figure 8: **The challenge of object localization when the model is conditioned solely on a textual prompt**. Text Prompt (e.g., "two red barns") is easier to cause a poor or incorrect layer extraction due to the model's inability to precisely locate the target object from semantic information alone. Conversely, providing a simple Spatial Prompt (e.g., a box) instantly resolves the ambiguity, guiding the model to produce a high-fidelity RGBA layer that accurately isolates the target object(s).

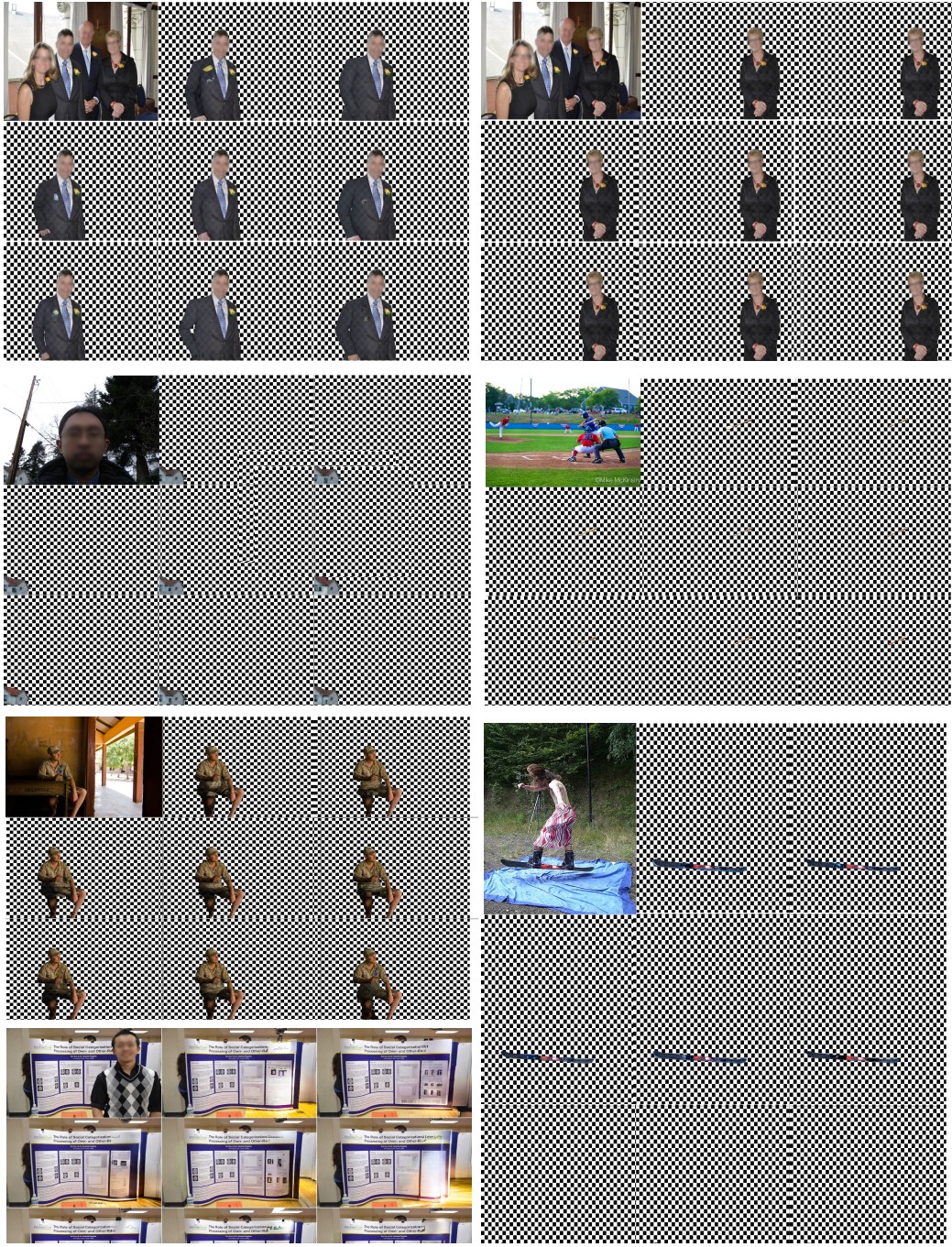

Figure 9: **RefLayer model qualitative result.** For each testing image (top-left corner of each block), we conduct the RefLayer model eight times. The generated result shows a diversity in completion, while preserving the visible content robustly.

### E.2   REFLAYER MULTI-GENERATION WITH DIFFERENT SEEDS.

Fig. 9 presents the result from the RefLayer model pretrained with the 1M training set and fine-tuned with 100K quality-tuning set. We verify that RefLayer has strong ability in prompt following, and has achieved a delightful ability in completing occluded region. We also notice that it still suffers from serious occlusion, unclear prompting (like point only), or small object decomposition.

### E.3 RefLayer v.s. Nano Banana Pro

We further experimented with Nano Banana Pro (Gemini 3). As shown in Fig. 10 Nano Banana Pro fails to perform satisfying layer decomposition. Despite prompting attempts, 1) it fails to preserve the original visible regions, especially for small object layers, but tends to generate new appearances; 2) for the large object layer, the localization is correct, but completing the occluded region remains a challenge; 3) it cannot generate real RGBA images, producing a fake checkerboard background instead. This suggests that current general-purpose generative LMMs do not yet possess the capability required for RLD, emphasizing the need for specialized datasets and models such as ours.

### E.4 RefLade Scene and Style

The RefLade dataset is constructed to maximize visual diversity, featuring a vast majority of Real Images (95%) and a portion of Stylized Images (5%). A visualization is shown in the Fig. 11.

### E.5 RefLayer Trained on MuLAN v.s. on RefLade

Besides the quantitative result in Tab. 6, Fig. 12 provides a qualitative comparison of RefLayer trained on MuLAN and on RefLade. The model trained on MuLAN generally produces lower-quality outputs—both in RGB appearance and alpha transparency compared to that trained on our RefLade dataset.

### E.6 Additional Visualization Result

Fig. 13 provides more qualitative results of the RefLayer model with respect to diverse prompts.

## F Large Language Models Acknowledgments

Besides the utilization of large language models (LLMs) in data engine (Appendix A.3), we acknowledge the use of LLM, specifically OpenAI's ChatGPT, to aid in improving the clarity the paper. The model was not used for research ideation, content generation, data analysis, or experimental design. All intellectual contributions remain those of the listed authors.

**Original Image**  **Google Gemini 3**
(Nano Banana Pro)  **RefLayer (Ours)**

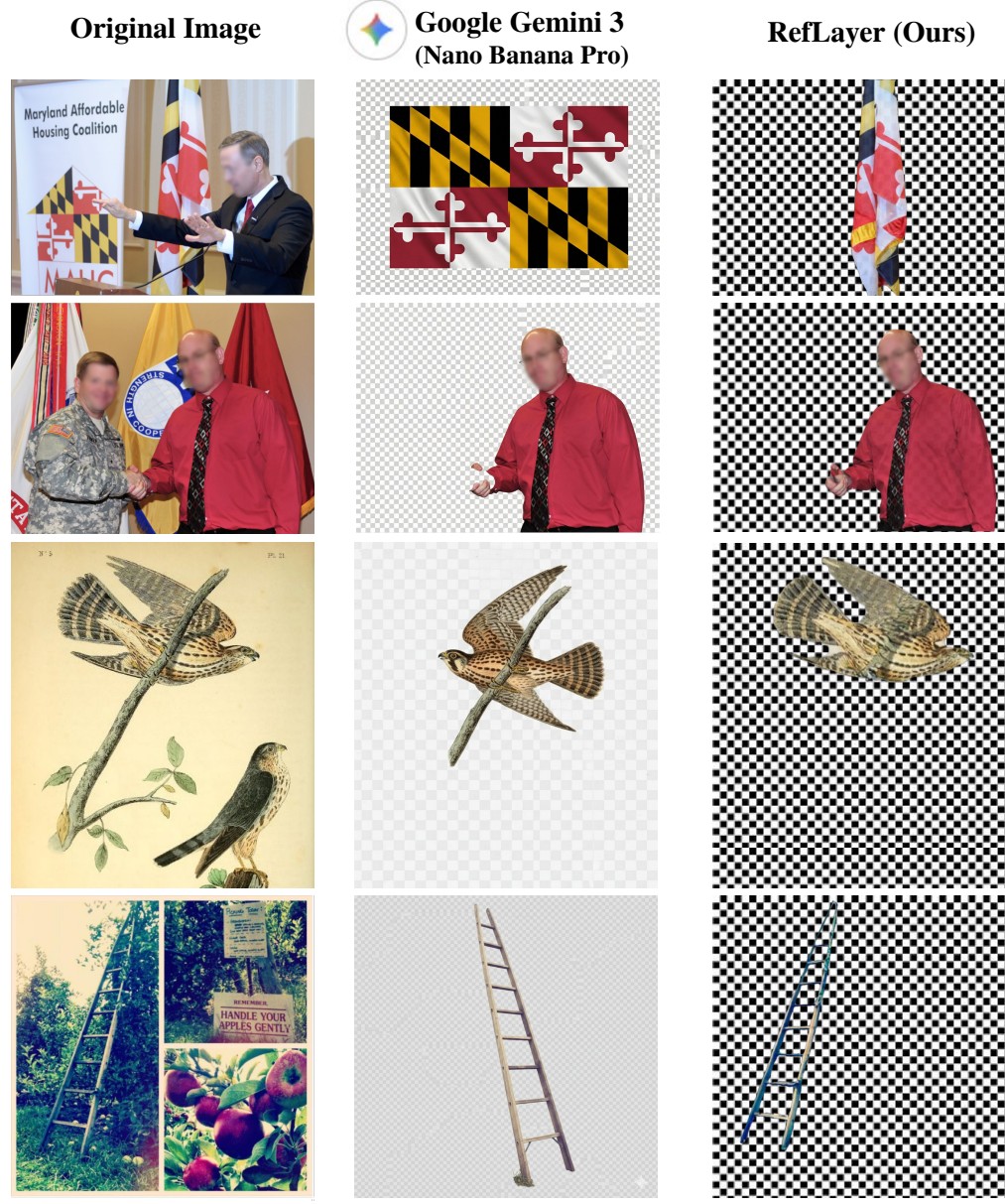

**Prompt used for Gemini 3 (Nano Banana Pro)**

*"You are an image editing expert. Given the input image, please perform the image layer decomposition task, where you need to extract the background and foreground layers as RGBA images. Each layer should either contain the background or one foreground object. When part of the layer is occluded by other layers, you need to complete the occluded area, such that the layer looks like a complete object without occlusion. You need to make sure the non-occluded area of layers are identical to the input image. Do not outpaint any content outside the original image boundary. Now, please generate the flag's complete layer image."*

Figure 10: **Limitations of existing general-purpose LMMs on the Referring Layer Decomposition (RLD) task**. We observe that Google Gemini 3 (Nano Banana Pro) struggles to preserve object identity, often regenerating the appearance entirely, and fails to produce valid RGBA outputs.

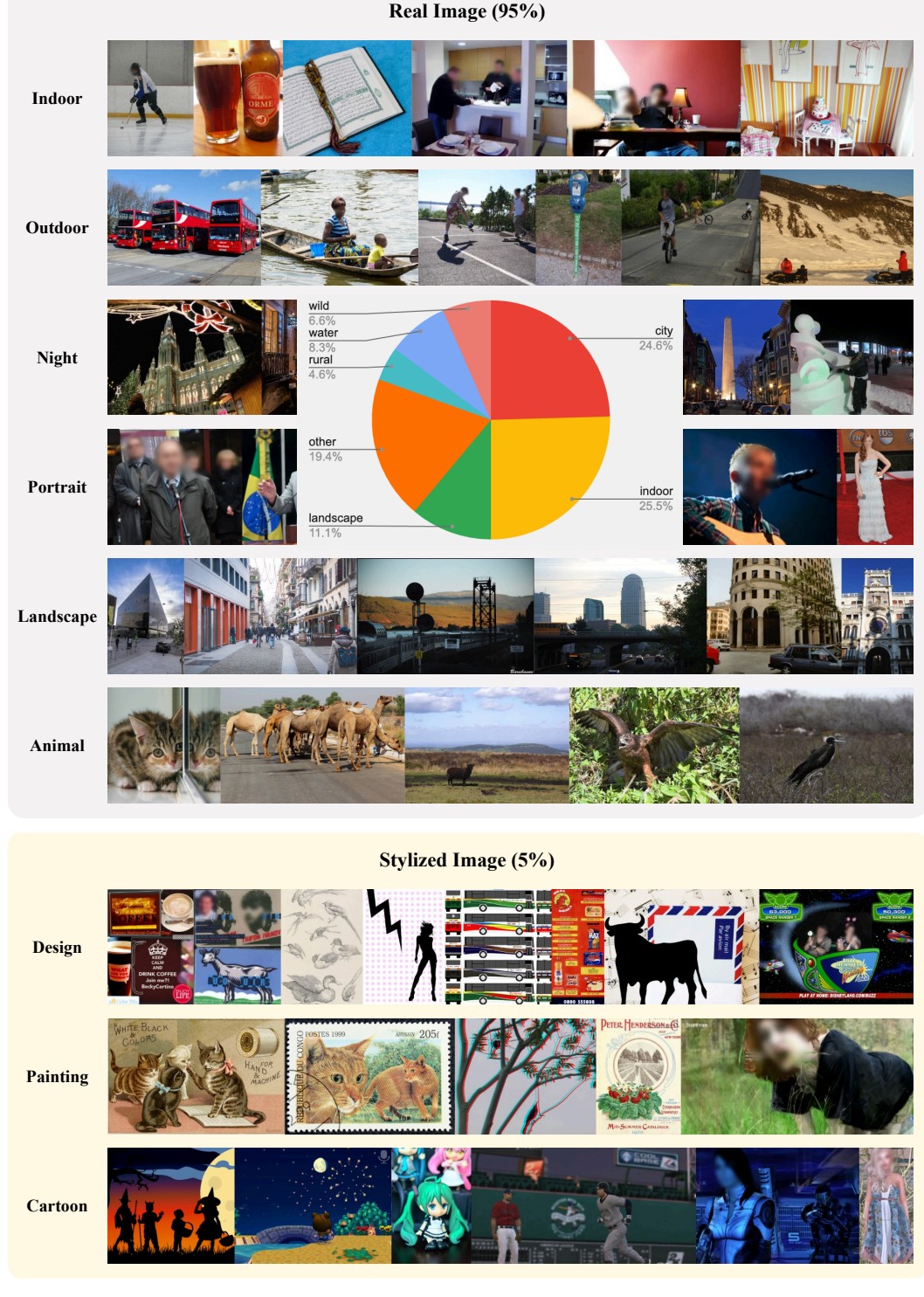

Figure 11: **Comprehensive Visual Diversity and Scene Category Distribution of the RefLade Dataset.** The figure illustrates the dataset's dual composition of 95% Real Images and 5% Stylized Images across nine distinct visual categories. The central pie chart quantifies the distribution of dominant scene types, showing strong coverage of Indoor (25.5%) and City (24.6%) scenes to ensure robust generalization across various photographic contexts.

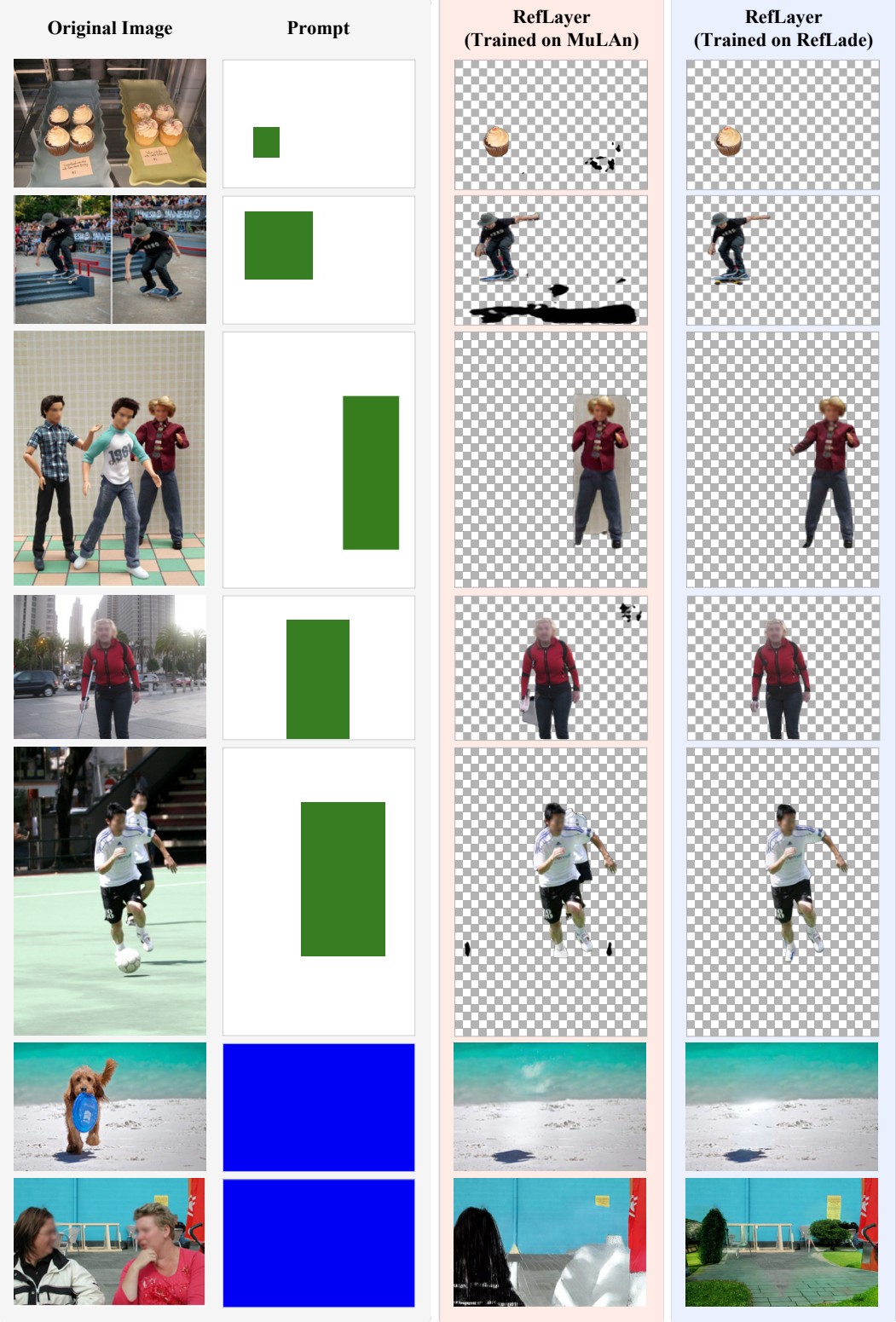

Figure 12: **Comparison between RefLayer trained on the MuLAn dataset and the same model trained on our RefLade dataset.** We observe that the results on the MuLAn dataset are generally worse than those obtained when training on our RefLade dataset, showing lower accuracy in the RGB content and alpha channel along object boundaries.

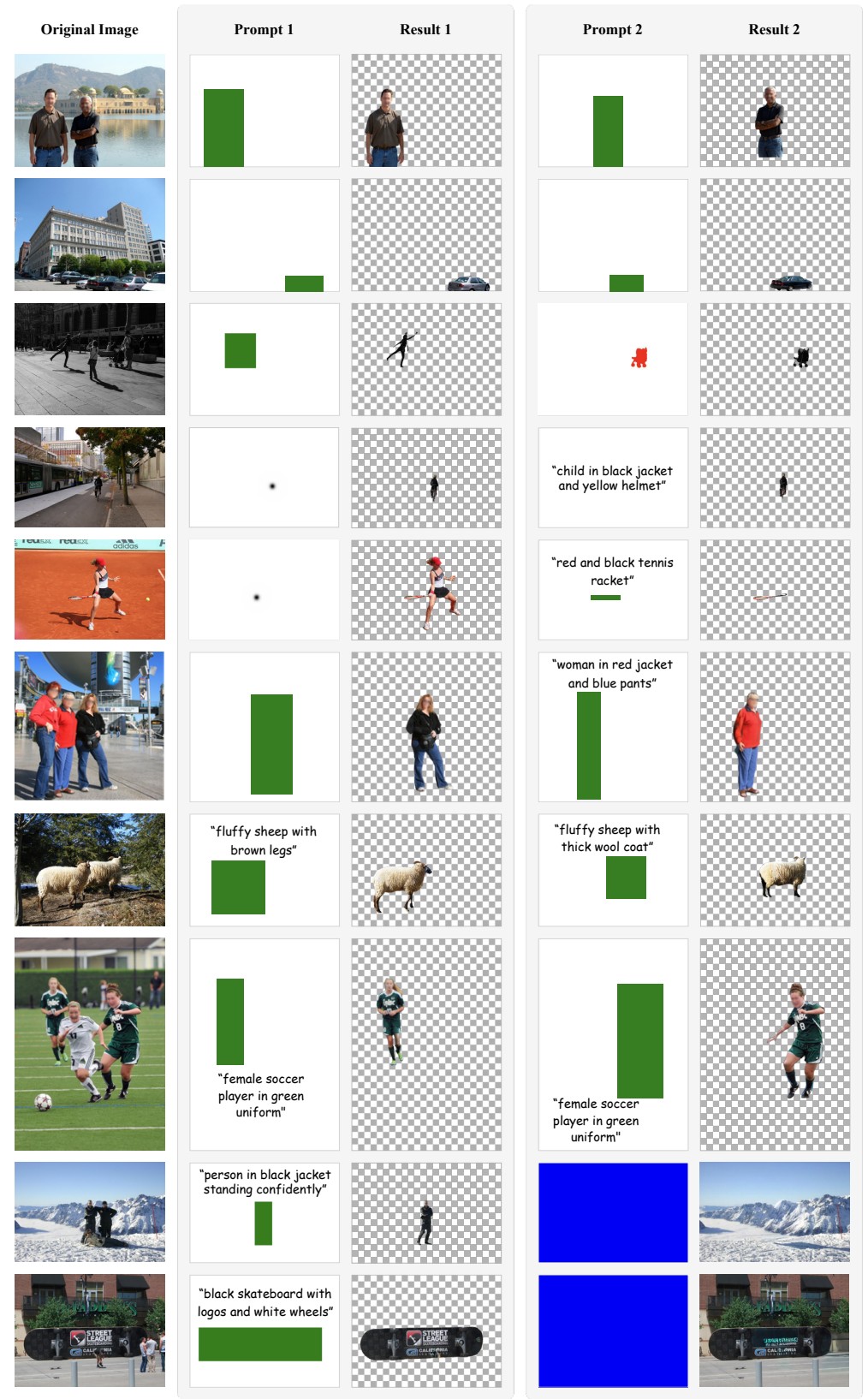

Figure 13: **More qualitative results of our RefLayer model with respect to diverse prompts.** Each row shows two different prompts and the corresponding results for the same input image.

