# OpenReview forum: "Referring Layer Decomposition"
_ICLR.cc/2026/Conference — ICLR 2026 Poster_

### Official Review · Reviewer_wYda · 2025-10-29

**Soundness:** 3
**Presentation:** 3
**Contribution:** 3
**Rating:** 4
**Confidence:** 3

**Summary:**

This paper introduces a new task that extracts a target RGBA layer guided by a user-provided prompt. To support this task, the authors present RefLade, a dataset comprising 1.11 million image–layer–prompt triplets. Building on RefLade, they establish an evaluation protocol along three key dimensions: preservation, completion, and faithfulness. Furthermore, the paper proposes RefLayer as a baseline model for this task. RefLayer encodes spatial prompts using color-coded maps integrated into the latent space and employs a parallel alpha decoder to generate complete object RGBA layers.

**Strengths:**

1. This paper is well-written and easy to understand.

2. This paper defines a novel task and presents a comprehensive framework to address it, including a dedicated dataset, evaluation metrics, and a baseline model.

**Weaknesses:**

1. In Figure 1, the flag extracted exhibits a noticeable color discrepancy (or color cast) when compared to the original image. The color shift is quite apparent and detracts from the quality of the result. The authors should investigate the cause of this artifact.

2. The paper's qualitative evaluation is currently a weak point. First, the paper lacks direct qualitative comparisons of different methods and ablation study. Without side-by-side visual comparisons, it is difficult for the reader to verify the claimed advantages of the proposed model over existing work. Second, the number of generated results shown is limited. The authors should include a more comprehensive and diverse set of examples (either in the main paper or, more extensively, in the supplementary material) to better demonstrate the model's capabilities, robustness, and potential failure cases.

**Questions:**

1. The authors employed a third-party, closed-source LLM to construct a large-scale dataset. Have the authors considered making this dataset publicly available? Since the primary contribution of this work lies in the dataset’s construction, the overall impact of the paper would be considerably limited if the dataset remains closed-source.

2. In the constructed dataset, approximately 25% of the training data contain errors. As mentioned in the appendix, these issues primarily result from failed restoration and segmentation inaccuracies. Have the authors conducted an analysis of these erroneous samples? Furthermore, do these problematic data limit the potential applications of the dataset? Do the authors plan to address these issues in future work?

---

> ### Author Response · Authors · 2025-11-22
>
> We thank the reviewer for recognizing our work as well written, easy to understand, and for acknowledging the novelty of the task and the comprehensiveness of our framework.
>
> **1. Cause of color discrepancy in Figure 1**
>
> We appreciate the reviewer's keen eye on identifying the color discrepancy in the flag example in Fig 1.  We investigated the reason and determined that this artifact stems from transparency prediction rather than a shift in color generation, i.e., caused by the RefLayer's alpha decoder.
>
> As we know, a layer consists of RGB color content and alpha layer for transparency.  We tried to remove the flag's alpha channel, then, the color looks identical to the original image. That means the model conducted properly on color prediction, while the alpha decoder thinks the flag is “semi-transparent” and predicts values around 0.9.
>
> This phenomenon suggests the current alpha-decoder needs enhancement. One alternative strategy in future work could be incorporating alpha calibration operation.  Since RefLayer is just a simple and the first baseline to this novel task, our primary objective is to establish a trainable, benchmarkable, and researchable foundation rather than to present a fully solved solution at this early stage.
>
> **2. Qualitative evaluation is currently a weak point (Direct qualitative comparisons and additional generated result)**
>
> We appreciate the reviewer wYda's great suggestion.  We have added the following figure for qualitative visualization and evaluation:
>
> 1) Fig. 10 Qualitative result illustration. RefLayer v.s. Gemini 3 (Nano Banana Pro);
>
> 2) Fig. 12 Qualitative result illustration. RefLayer trained on MuLAn vs trained on RefLade;
>
> 3) Fig. 13  Qualitative result illustration.  More result from RefLayer as Fig. 1's extension.
>
> **3. Dataset accessibility**
>
> We will make it publicly available.
>
> **4. Do these problematic data limit the potential applications of the dataset? Do the authors plan to address these issues in future work?**
>
> In Appendix A.4, we analyzed that inpainting errors account for ~65% of failures (e.g., hallucinations or artifacts in occluded regions), and segmentation errors account for ~20% (e.g., incomplete masks from SAM). While ~25% of the data in the pre-training set contains noise (rated “neutral” or “poor” in our audit), we argue that this does not limit the dataset's usefulness.
>
> Firstly, large-scale human cleaning on pretraining data is not cost-efficient.  Therefore, we additionally provided 100K high quality human-cleaned data for finetuning. Secondly, our experiments verify that RefLayer trained on the noisy 1M dataset (RefLade) and finetuned on the 100K high-quality dataset significantly outperforms models trained only on the 100K high-quality datasets, suggesting the noise is manageable and does not impact the overall usability of the pretraining dataset.
>
> With the rapid development of deep learning, we will continuously upgrade the underlying foundation models as they advance, which will naturally reduce error rates in the data engine pipeline.

---

> > ### Comment · Reviewer_wYda · 2025-11-27
> >
> > The authors' response has addressed my concerns. After considering the comments from other reviewers, I have raised my score.

---

### Official Review · Reviewer_rYGP · 2025-11-01

**Soundness:** 3
**Presentation:** 2
**Contribution:** 3
**Rating:** 6
**Confidence:** 2

**Summary:**

This paper introduce the Referring Layer Decomposition (RLD) task, a new task which predicts
complete RGBA layers from a single RGB image, conditioned on flexible
user prompts.  a large-scale
dataset (RefLade) comprising 1.11M image–layer–prompt triplets  is constructed.  In addition, RefLayer is proposed as a simple baseline for
prompt-conditioned layer decomposition.

**Strengths:**

1. This paper introduces Referring Layer Decomposition (RLD), the pioneering task that explores layer decomposition guided by multi-modal referring inputs.

2. The authors introduce RefLade, a large-scale dataset of 1.11 million image-layer-prompt triplets built using a scalable data engine. With its human-curated splits for tuning and testing and a well-defined evaluation protocol, RefLade facilitates and paves the way for future RLD studies. RefLayer is also desigend as a simple baseline.

**Weaknesses:**

1. More details to ensure the correctness of the image–layer–prompt triplets should be given. In scene understanding, the availabel models for object detection and instance-segmentation can not perform well in all situations, especially for small or occlude objects. How doauthors deal with these cases?

2. It would have been better to show the image distribution with respect to styles, e.g., real images, cartoon， posters and so on, and discuss the model performance in different styles.

**Questions:**

What are the computational costs in terms of both human effort and GPU resources  to construct  training and testing datasets.

---

> ### Author Response · Authors · 2025-11-22
>
> We thank the reviewer for recognizing our work as pioneering task,  well-defined evaluation protocol,  and facilitates and paves the way for future RLD studies.
>
> **1. Improve the correctness of the image–layer–prompt triplets**
>
> We implemented a multi-stage strategy within our Data Engine (Section 3.1 and Appendix A.1). This strategy focuses on filtering out failure-prone inputs, ensembling detection/segmentation models for robustness, and validating outputs via quality controls.
>
> 1. Pre-filtering. We proactively mitigate the reliance on models handling extreme cases by filtering out images where failure is highly probable before processing begins, where one constraint empirically defines if an image contains more than 10 objects AND their average mutual Intersection over Union (IoU) exceeds 0.2, the image is discarded. This ensures we avoid highly clustered scenes that remain intractable for current technologies.
>
> 2. Ensembling. Ensembling detection and segmentation rather than relying on a single model.
>
> - Detection Ensemble: We combine closed-set detection (RT-DETR), open-vocabulary detection (OWL-V2), and MLLM-based grounding (GPT-4o + Grounding-DINO). Our experiments in Appendix A.2 show this ensemble achieves 0.652 precision/0.659 recall, significantly outperforming individual models like OWL-V2 (0.236/0.621).
>
> - Segmentation Ensemble: we ensemble instance segmentation with panoptic segmentation (OpenSeeD) to refine object masks by excluding background masks.
>
> 3. Detail Preservation: For small objects/low-res images, we employ super-resolution model prior to processing to enhance detail and improve segmentation performance.
>
> 4. Post-Filtering. We employ an automatic and rigorous post-filtering stage that evaluates the generated triplets on three dimensions:
> - Visual Quality: We use Gemini to score the generated RGBA layers on structure, edge quality, and realism.
>
> - Semantic Consistency: We calculate the CLIP similarity between the generated layer and the prompt to ensure the object was correctly identified.
>
> - Preservation: We verify that the visible region of the generated layer aligns pixel-wise with the original image to prevent hallucination or drift.
>
> 5. Human Validation. The high-quality tuning set and testing set are validated and cleaned by human annotators.
>
> **2. Show the image distribution with respect to styles**
>
> To examine the image distribution with respect to style, we randomly sampled 600 images from the RefLade dataset and annotated them manually. The results show that among the sampled images, 95% are real images, and 5% belong to stylized images.  We also show the scene distribution and visualize some examples in Appendix Fig. 11.
>
> **3. Computational costs in terms of both human effort and GPU resources**
>
> We list the cost in the table below.
>
> | | Cost |
> | :--- | :--- |
> | **Human Cleaning** | ~ 1K layers per annotator per day. |
> | **GPU resources for data pipeline** | ~ 0.05 gpu hour per layer (80GB A100). |
> | **Gemini** | ~ 2K input tokens and 1K output tokens per layer. |
> | **GPT** | ~ 1K input tokens and 0.5K output tokens per layer. |

---

### Official Review · Reviewer_HNvP · 2025-11-10

**Soundness:** 3
**Presentation:** 3
**Contribution:** 3
**Rating:** 8
**Confidence:** 3

**Summary:**

This paper proposes a new task called Referring Layer Decomposition (RLD). RLD is conceptually similar to referring image segmentation, whose inputs can be text or spatial cues of the target object. The output of the task is the layer representation of the target object (RGBA, completed object + alpha mask). Since the most critical factor for generative AI models is the data, this work scales up the RGBA data generation pipeline and creates the RefLade dataset which is significantly larger than prior real-world RGBA datasets. Then, authors design an evaluation metric, which is shown to be aligned with human preference. Finally, a simple baseline model, RefLayer, is designed and fine-tuned on the proposed dataset. Despite its simple design, it achieves significantly better performance than prior works.

**Strengths:**

- The paper is well-written and easy to follow. The figures and charts help understand the pipeline and the statistics of the dataset very well.
- We all know that data is key to the advance of GenAI. This paper proposes a large-scale, real-world dataset for an interesting task. The data generation pipeline involves several SOTA models which guarantee the high quality of the dataset.
- The analysis on data scale, data quality (different subsets), and pre-trained models is comprehensive.
- I love how the proposed metric is aligned with human preference. This is critical for a useful metric to monitor real model progress.

**Weaknesses:**

I do not see any major weaknesses in the paper. As a paper that defines a new task and proposes a new dataset, every aspect of it is executed very well. Maybe one concern is having more baselines: the paper proposes its own simple baseline which is great, but would it be possible to adapt some prior work's models to this task and benchmark against your proposed method?

Another weakness is, can you show some downstream application of the dataset, similar to Sec.4.4 of the MULAN paper. For example, is it possible to fine-tune LayerDiffuse on the single-object subset of RefLade, and see if that improves performance? Or fine-tune InstructPix2Pix to perform object removal & insertion tasks? These results will definitely make the paper stronger, though I think the current form is already enough for acceptance.

**Questions:**

- The HPA scores are low when using text prompts. Can you provide some qualitative failure case analysis? One issue might be there are multiple similar objects (e.g. belong to the same category) in an image, so text prompts along cannot disambiguate between them.
- What are model 1-9 in Fig.4? Are they the same model trained on different steps / amount of data?
- Typo in Fig.1 "Linguistic Prompting" part: “The Brown and white house” should be "horse” not "house".

---

> ### Author Response · Authors · 2025-11-22
>
> We thank the reviewer for recognizing our large-scale, real-world dataset, interesting task, comprehensive analysis, and useful metric.
>
> **1. Having more baselines, adapt some prior work models to this task and benchmark against your proposed method.**
>
>   RLD is a newly defined task and there are rare existing open-source methods that can be directly applied to referring layer decomposition. Diffusion-based generative/editing models are good candidates for adaptation if extended with an RGBA-aware decoding head, and we have included several strong generative/editing models (InstructPix2Pix, SD3, and UltraEdit) in Table 5 to illustrate how different models behave when applied to this task.
>
> **Table 5: Ablation study on Base Model**
>
> | Base Model | Canvas | HPA_frgd | HPA_occ |
> | :--- | :--- | :--- | :--- |
> | SD3 | Board | 0.4275 | 0.3692 |
> | InstP2P | Board | 0.4534 | 0.4100 |
> | UltraEdit | Board | **0.4698** | **0.4187** |
>
>   Additionally, we have presented Appendix D.3, where Pix2Gestalt, a recent work on amodal segmentation and completion, is zero-shot benchmarked. Pix2Gestalt is trained on its own synthetic data. As shown in Table 7, RefLayer outperforms Pix2Gestalt in terms of all metrics.
>
> **Table 7: Comparison with amodal completion methods on RefLade test subset.**
>
> | Method | HPA_occ ↑ | Foreground HPA ↑ | Foreground FID ↓ | Foreground LPIPS ↓ | Foreground DIR ↑ |
> | :--- | :--- | :--- | :--- | :--- | :--- |
> | Pix2Gestalt (Ozguroglu et al., 2024) | 0.2687 | 0.3397 | 18.23 | 0.1671 | 0.1198 |
> | MuLAn (Tudosiu et al., 2024) | 0.3041 | 0.3852 | 22.68 | 0.1403 | 0.2031 |
> | RefLaye | **0.4403** | **0.4833** | **10.43** | **0.1298** | **0.2586** |
>
>   We further experimented the very recent Nano Banana Pro (Gemini 3).  It fails to perform satisfying layer decomposition.  Despite extended prompting attempts, 1) it fails to preserve the original visible regions, especially for small object layers, but tends to generate new appearances; 2) for large object layers, the localization is correct, but completing occluded region remains a challenge; 3) it cannot generate real RGBA images, producing a fake checkerboard background instead. This suggests that current general-purpose generative LMMs lack the capabilities required for RLD, emphasizing the importance of specialized datasets and models like ours. See Appendix E.3 and Fig. 10.
>
> **2. Downstream application of the dataset.**
>
> In Appendix C.1, we discussed potential applications in real-world products. Specially,
>
> 1. Full-Scene Decomposition Agent
>
> 2. Scalable RGBA Image Generation
>
> 3. Seamless Integration with Existing Tasks
>
> In Appendix D.2, we applied the model trained on RefLade to the downstream task of Amodal Segmentation on the standard COCOA benchmark.  Despite never seeing COCOA data, our model achieved state-of-the-art performance on the miou-occ metric (38.83 vs. 27.71 for previous SOTA), significantly outperforming fully supervised methods.
>
> We agree with the reviewer that fine-tuning on our datasets for object removal & insertion tasks is a good downstream application, and we include this as future work and add this discussion to Appendix C.1.
>
> **3. The HPA scores are low when using text prompts.**
>
> We appreciate the reviewer's insightful observation. We have also noticed this phenomenon.  The major reason is the inherent difficulty of text-only prompting generation mode.
>
> We visualize this challenge in Fig. 8. The text-only prompt inherently requires the model to perform an additional reasoning and localization step to correctly identify the target object. For instance, given the prompt “two red barns” (top row), the model incorrectly extracts a large portion of the foreground ground plane and horses, demonstrating a failure to isolate the target based on the language alone. In contrast, a simple Spatial Prompt (e.g., a box) provides an explicit localization cue, guiding the model to accurately decompose the target object(s). The prompt “person standing in the middle” (bottom row) similarly yields a poor result without spatial guidance, but a generic box yields a high-quality layer extraction.
>
> A promising approach to bridge this gap could involve applying a grounding model to provide additional spatial prompting cues when running RefLayer in text-only mode. Alternatively, a unified and more powerful initialization, possessing strong localization and reasoning ability, could also significantly improve performance.
>
> We added this analysis in Appendix E.1.
>
> **4. What are models 1-9 in Fig.4?**
>
> In Fig. 4, models 1–9 (A–I) differ in their underlying base models (InstructP2P, SD3, UltraEdit), the amount of training data used, and the number of training epochs. These variations ensure that the models cover a wide performance range. All models remain anonymous during the ELO evaluation.
>
> **5. Typo in Fig.1**
>
> Thanks for pointing it out. We have updated accordingly.

---

> > ### Comment · Reviewer_HNvP · 2025-11-22
> >
> > I thank the authors for the rebuttal. My concerns are addressed. After reading the review from other reviewers, I decided to keep my score.

---

### Official Review · Reviewer_kTb4 · 2025-11-10

**Soundness:** 4
**Presentation:** 4
**Contribution:** 3
**Rating:** 8
**Confidence:** 3

**Summary:**

The paper introduces the task of Referring Layer Decomposition (RLD), which aims to recover a complete RGBA layer corresponding to a user-provided prompt, such as spatial maps or text. To support this task, the authors construct RefLade, a large-scale dataset of image-layer-prompt triplets produced through a combination of automatic generation and human curation, along with an evaluation protocol assessing preservation, completion, and faithfulness. A diffusion-based baseline, RefLayer, is proposed and evaluated extensively. Overall, the paper presents a well-motivated and clearly defined formulation, a high-quality dataset, and a comprehensive baseline, offering a valuable foundation for future research on image layer decomposition.

**Strengths:**

- The proposed dataset constitutes a significant improvement over existing resources for this problem domain, both in scale and in the level of curation. The combination of automated and manual verification enhances its overall quality and reliability.
- The paper provides thorough evaluations, including analyses of design choices, as well as assessments of the alignment between the proposed metrics and human judgments.
- The paper is clearly written, with a well-motivated problem statement and sufficient technical and implementation details.

**Weaknesses:**

The paper is overall good, and I didn't find major weaknesses. One minor:

For the completion metric, what is the rationale for defining it as the difference between CLIP embeddings, $f(g_\text{rgb}) - f(g_\text{rgb} * g_v)$, rather than directly using the CLIP embedding of the non-visible region, $f(g_\text{rgb} * (1 - g_v))$?
The motivation for this specific formulation should be clarified.

**Questions:**

No questions

---

> ### Author Response · Authors · 2025-11-22
>
> We thank the reviewer for recognizing our work as well motivated, clearly defined, and significant improvement.
>
> **Completion metric**
>
> The concept behind our Image Directional Similarity metric is inspired by the widely adopted CLIP text-image directional similarity, which has been used extensively in image editing tasks such as [R1, R2, R3, R4]. This metric evaluates how well an image transformation aligns with a given text instruction. It does so by first defining a semantic “direction” in CLIP's text embedding space—calculated as the vector difference between two prompts (e.g., “a photograph” vs. “a sketch”)—and then comparing this with the direction in image embedding space between the original and edited images. The similarity between these two directions, measured by cosine similarity, reflects how faithfully the image follows the instruction.
>
> Our proposed Image Directional Similarity metric adapts this idea to the RLD task. Instead of measuring changes from textual instructions, we define the direction as the vector difference between the visible parts of an object and its fully completed, ground-truth RGBA layer. The metric then evaluates whether the predicted layer follows a similar trajectory—essentially asking: Does the generated content align with the "completion action" in a way that parallels the ground-truth?
>
> We also thank the reviewer kTb4 proposes the alternative formulation using the CLIP embedding of only the non-visible region (1 − gv).  In RLD tasks, focusing only on the non-visible regions is often not semantically reliable. For instance without losing generality, in Appendix Fig. 9, the man in suit is occluded by the woman, and his arm is invisible region.  If we only evaluate the non-visible region (1 - gv), we focus on the texture and shape difference of the completed region--which is not uniquely defined (all eight predictions appear reasonable). Instead, the  Image Directional Similarity metric evaluates within the context of the entire object from a global view,  capturing the high-level semantic information needed to make the man whole.
>
> [R1] Tim Brooks, Aleksander Holynski, and Alexei A Efros. Instructpix2pix: Learning to follow image editing instructions. In CVPR, 2023.
>
> [R2] Yuzhou Huang, Liangbin Xie, Xintao Wang, Ziyang Yuan, Xiaodong Cun, Yixiao Ge, Jiantao Zhou, Chao Dong, Rui Huang, Ruimao Zhang, et al. Smartedit: Exploring complex instruction-based image editing with multimodal large language models. In CVPR, 2024.
>
> [R3] Jun Zhou, Jiahao Li, Zunnan Xu, Hanhui Li, Yiji Cheng, Fa-Ting Hong, Qin Lin, Qinglin Lu, and Xiaodan Liang. Fireedit: Fine-grained instruction-based image editing via region-aware vision language model, 2025.
>
> [R4] Lvmin Zhang, Anyi Rao, and Maneesh Agrawala. Adding conditional control to text-to-image diffusion models, 2023.

---

### Author Response · Authors · 2025-11-22
**To all reviewers**

We sincerely thank all the reviewers for their valuable comments and suggestions. We have updated the manuscript accordingly, and new content in the revised version is marked with the label “NEW” for easy reference.

---

### Author Response · Authors · 2025-12-03
**Summary of the Rebuttal Process**

We thank all reviewers and ACs for the time and effort in reviewing our paper and the contributions to the community. We are providing a summary of our paper and the rebuttal process to help present a clearer overview of our paper.

Our work was **initially scored 8,8,6,4.**  All reviewers appreciate that RLD is a clearly defined, well-motivated, and meaningful new task.  Notably, reviewers kTb4 and HNvP find **no major weakness** in our paper. After the first round discussion, reviewer wYda **raises his/her score to 6**; reviewer HNvP expressed satisfaction, stating their concern has been addressed.

Our rebuttal addressed key concerns, which we summarize below:

**1. Baseline and prior work models (HNvP)**

We clarified our baseline experiments with SD3, InstructPix2Pix, UltraEdit, as well as additional experiments in the Appendix with Pix2Gestalt and MuLAn. We further evaluated the recent Nano Banana Pro (Gemini 3) on RLD, compared it against our model, and analyzed its limitations and failure cases.

**2. The HPA scores are low when using text prompts (HNvP)**

We provide additional analysis explaining why HPA scores are lower when using text prompts, as this setting requires the model to perform additional reasoning and localization, making the task inherently more challenging.

**3. Clarification on quality assurance, dataset image distribution, and  Computational costs (rYGP, wYda)**

We give a detailed explanation and clarification on how to improve the image-layer-prompt triplet. We also present an image distribution and computational cost for the reviewer's reference.

**4. Qualitative evaluation (wYda)**

We have added 3 additional figures in the appendix for qualitative visualization and evaluation, which satisfy the reviewer.

**5. Others**

We have also provided point-to-point responses to the concerns about the rationale for using a directional formulation (kTb4), the Cause of color discrepancy in Figure 1 (wYda), and updated the submission accordingly.

We have carefully addressed all reviewer comments and revised the submission accordingly. We believe these updates further strengthen the contribution and clarity of our work.

We sincerely thank the reviewers, ACs, and PCs for their thoughtful engagement and valuable feedback.

---

### Meta-Review · Area_Chair_4CTm · 2026-01-07

**Summary:**

This paper received overall positive feedback from the reviewers. The paper presents a novel tasks by proposing a new dataset and benchmark, the details are clearly presented, extensively evaluated, and the application itself is very practical and useful. The qualitative results requested by one of the reviewers have been added to the revised version and look convincing. AC is glad to recommend acceptance. The details are below.

**Reviewer Concerns:**

- Reviewer kTb4 concerns / questions on the metric formulation has been clarified and explained by the authors.
- Reviewer HNvP has concerns on the other baselines which have been made up. Authors also show the visual of text-guided extraction to illustrate the failure cases causing low HPA scores. AC believes the concerns have been addressed.
- Reviewer rYGP requests the model details and the image distribution, which have been made up in the revised version. Stylized image occupies a small ratio though, which might not be good enough for a balanced dataset.
-  Reviewer wYda noticed that the color issues exist in the results, and the authors admit it was caused by alpha channel of the RGBA decoding, and plan to improve it later. The authors also add more qualitative visualization.

**Reviewer Scores:**

Reviewer kTb4, HNvP and ostu will possibly remain the same scores or higher, and Reviewer wYda will increase the score given the concerns are addressed.

---

### Decision · Program_Chairs · 2026-01-26

Accept (Poster)